# BCAS2 promotes primitive hematopoiesis by sequestering β-catenin within the nucleus

**Guozhu Ning[1,2†], Yu Lin[3†], Haixia Ma[4,5], Jiaqi Zhang[4,5], Liping Yang[1], Zhengyu Liu[1], Lei Li[4,5]\*, Xinyu He[1]\*, Qiang Wang[1]\***

[1]Innovation Centre of Ministry of Education for Development and Diseases, the Sixth Affiliated Hospital, School of Medicine, South China University of Technology, Guangzhou, China; [2]Affiliated Hospital of Guangdong Medical University & Key Laboratory of Zebrafish Model for Development and Disease of Guangdong Medical University, Zhanjiang, China; [3]School of Pharmacy, Qiqihar Medical University, Qiqihar, China; [4]Institute State Key Laboratory of Stem Cell and Reproductive Biology, Institute of Stem Cell and Regeneration, Beijing Institute of Stem Cell and Regenerative Medicine, Institute of Zoology, Chinese Academy of Sciences, Beijing, China; [5]University of Chinese Academy of Sciences, Beijing, China

**\*For correspondence:**
lil@ioz.ac.cn (LL);
hexyu@scut.edu.cn (XH);
qiangwang@scut.edu.cn (QW)

[†]These authors contributed equally to this work

**Competing interest:** The authors declare that no competing interests exist.

## eLife Assessment

This **important** work supports the role of breast carcinoma amplified sequence 2 (Bcas2) in positively regulating primitive wave hematopoiesis through amplification of beta-catenin-dependent (canonical) Wnt signaling. The study is **convincing**: it uses appropriate and validated methodology in line with the current state-of-the-art, and there is a first-rate analysis of a strong phenotype with highly supportive mechanistic data. The findings shed light on the controversial question of whether, when, and how canonical Wnt signaling may be involved in hematopoietic development. The work will be of interest to hematologists and developmental biologists.

**Abstract** Breast carcinoma amplified sequence 2 (BCAS2), a core component of the hPrP19 complex, plays crucial roles in various physiological and pathological processes. However, whether BCAS2 has functions other than being a key RNA-splicing regulator within the nucleus remains unknown. Here, we show that BCAS2 is essential for primitive hematopoiesis in zebrafish and mouse embryos. The activation of Wnt/β-catenin signaling, which is required for hematopoietic progenitor differentiation, is significantly decreased upon depletion of *bcas2* in zebrafish embryos and mouse embryonic fibroblasts. Interestingly, BCAS2 deficiency has no obvious impact on the splicing efficiency of β-catenin pre-mRNA, while significantly attenuating β-catenin nuclear accumulation. Moreover, we find that BCAS2 directly binds to β-catenin via its coiled-coil domains, thereby sequestering β-catenin within the nucleus. Thus, our results uncover a previously unknown function of BCAS2 in promoting Wnt signaling by enhancing β-catenin nuclear retention during primitive hematopoiesis.

## Introduction

Hematopoiesis refers to the lifelong process by which all blood cell lineages are generated. It begins at the early stage of embryonic development, providing the growing embryo with sufficient oxygen and nutrients (***Galloway and Zon, 2003***). Evolutionarily conserved across vertebrate species,

hematopoiesis consists of two successive and partially overlapping waves: primitive and definitive. In mammals, the first wave of hematopoiesis occurs in the yolk-sac blood islands, producing primitive erythroid, megakaryocyte, and macrophage progenitors, which can be observed in mouse embryos as early as embryonic day 7.25 (E7.25) (*Murry and Keller, 2008*; *Ferkowicz and Yoder, 2005*; *Palis, 2016*). In zebrafish, primitive hematopoiesis initiates at around 11 hours post fertilization (hpf), when hemangioblasts emerge from the anterior lateral mesoderm (ALM) and posterior lateral mesoderm (PLM) and later differentiate into both hematopoietic and endothelial cells (*Paik and Zon, 2010*; *Detrich et al., 1995*; *Leung et al., 2005*).

Breast cancer amplified sequence 2 (BCAS2), also known as pre-mRNA splicing factor SPF27, is a 26 kDa nuclear protein containing two coiled-coil (CC) domains (*Kuo et al., 2009*). It was initially found to be overexpressed and amplified in human breast cancer cell lines (*Neubauer et al., 1998*; *Nagasaki et al., 1999*; *Qi et al., 2005*). Further studies have identified BCAS2 as a vital component of the human Prp19/CDC5L complex, which forms the catalytic ribonucleoprotein (RNP) core of spliceosome and is required for the activation of pre-mRNA splicing (*Neubauer et al., 1998*; *Ajuh et al., 2000*; *Grote et al., 2010*). In *Drosophila*, the function of BCAS2 in RNA splicing is essential for cell viability (*Chen et al., 2013*). In mouse, disruption of *Bcas2* in male germ cells impairs mRNA splicing and leads to a failure of spermatogenesis (*Liu et al., 2017*). Additionally, BCAS2 has been shown to be a negative regulator of p53 by directly interacting with p53 or modulating alternative splicing of *Mdm4*, a major p53 inhibitor (*Kuo et al., 2009*; *Yu et al., 2019*). Zebrafish *bcas2* transcripts were enriched in the sites of both primitive and definitive hematopoiesis during embryonic development (*Yu et al., 2019*). However, a previous study showed that p53 overactivation induced by zebrafish *bcas2* depletion did not affect primitive hematopoiesis, but impaired definitive hematopoiesis (*Yu et al., 2019*). In recent years, several studies have highlighted the importance of regulating the expression and activity of p53 in primitive erythroid cell differentiation in both mouse and zebrafish embryos (*Bissinger et al., 2018*; *Yang et al., 2023*; *Stanic et al., 2019*). Thus, it is necessary to reexamine the exact function of BCAS2 in primitive hematopoiesis.

Wnt signaling, usually categorized into canonical and non-canonical pathways, is involved in the process of hematopoiesis (*Richter et al., 2017*; *Krimpenfort and Nethe, 2021*; *Kokolus and Nemeth, 2010*). Notably, the canonical Wnt signaling pathway, which is dependent on the nuclear accumulation of β-catenin to regulate gene transcription, controls primitive hematopoietic progenitor formation and promotes definitive hematopoietic stem cell (HSC) specification (*Tarafdar et al., 2013*; *Nostro et al., 2008*; *Sturgeon et al., 2014*). For instance, it has been demonstrated in *Xenopus* that Wnt4-mediated activation of Wnt/β-catenin signaling plays a critical role in the induction and maintenance of primitive hematopoiesis (*Tran et al., 2010*). Moreover, transient inhibition of canonical Wnt signaling in zebrafish embryos impairs embryonic blood formation (*Lengerke et al., 2008*). However, previous studies utilizing human pluripotent stem cells revealed an opposite role of Wnt/β-catenin pathway in primitive progenitor generation (*Sturgeon et al., 2014*; *Paluru et al., 2014*). Therefore, the impact of Wnt/β-catenin signaling on primitive hematopoiesis remains elusive and even controversial. Moreover, it has been suggested that BCAS2 is important for neural stem cell proliferation and dendrite growth in mice by regulating β-catenin pre-mRNA splicing (*Chen et al., 2022*; *Huang et al., 2016*). As a nuclear protein, it is unclear whether BCAS2 can modulate Wnt/β-catenin signaling in a splicing-independent manner.

In this study, we generated two zebrafish *bcas2* mutant lines, both of which exhibited defects in male fertility and embryonic HSC formation, similar to what was previously reported in mice and zebrafish (*Liu et al., 2017*; *Yu et al., 2019*). More importantly, loss-of-function experiments suggest that BCAS2 is necessary for primitive hematopoiesis in both zebrafish and mouse embryos. We further find that *bcas2* is dispensable for the survival and proliferation of hematopoietic cells, but plays a crucial role in the differentiation of the hematopoietic lineage from hemangioblasts. Using a comprehensive approach, we reveal that BCAS2 is a nuclear retention factor for β-catenin during primitive hematopoiesis. Subsequent biochemical and functional experiments demonstrate that BCAS2 directly binds to β-catenin and suppresses its nuclear export to promote Wnt signal activation and hematopoietic progenitor differentiation. Furthermore, the CC domains on BCAS2 and the Armadillo (ARM) repeats on β-catenin are responsible for their interaction. Collectively, we have uncovered a novel function of BCAS2 in regulating Wnt/β-catenin signaling by sequestering β-catenin within the nucleus during primitive hematopoiesis.

# Results

## BCAS2 is necessary for primitive hematopoiesis

To confirm that *bcas2* is expressed in the posterior intermediate cell mass (ICM) where primitive hematopoiesis occurs in zebrafish, we first examined the spatiotemporal expression pattern of *bcas2* during zebrafish embryo development by performing whole-mount in situ hybridization (WISH). The results showed that *bcas2* was ubiquitously expressed from 1-cell stage to 10-somite stage (*Figure 1—figure supplement 1*). Its expression in the ICM became detectable at 18 hpf and was significantly elevated at 22 hpf (*Figure 1A*). We further observed that *bcas2* was co-expressed with the primitive erythropoietic marker *gata1* in the ICM at 22 hpf by fluorescence in situ hybridization (FISH) (*Figure 1B*). In contrast, *bcas2* was hardly detectable in the ICM in *cloche*$^{-/-}$ mutants that lack both endothelial and hematopoietic cells (*Figure 1C*). These results demonstrate a dynamic expression of *bcas2* in the ICM and imply a potential role of this gene in primitive hematopoiesis.

To gain insight into the developmental function of *bcas2*, we employed CRISPR/Cas9 system to generate *bcas2* mutants. Two mutant lines were obtained, designated *bcas2*$^{\Delta 7}$ (with a 7-base deletion) and *bcas2*$^{\Delta 14}$ (with a 14-base deletion). These mutations led to premature translation termination, which resulted in truncated Bcas2 proteins lacking the C-terminal CC domains (*Figure 1—figure supplement 2A and B*). *bcas2*$^{+/\Delta 7}$ and *bcas2*$^{+/\Delta 14}$ mutants were identified by T7 endonuclease I assay or restriction enzyme analysis (FspI) (*Figure 1—figure supplement 2C*). We found that nearly 85% of the embryos derived from crossing *bcas2*$^{+/-}$ males with *bcas2*$^{+/-}$ females did not develop to the cleavage stage (*Figure 1—figure supplement 2D*). Only 3% of the living embryos were homozygotes. In contrast, embryos obtained by crossing between wild-type males and *bcas2*$^{+/-}$ females were viable and showed normal morphology, with a heterozygosity rate consistent with Mendelian inheritance. This could be explained by male infertility as previously documented in *Bcas2* knockout mice (*Liu et al., 2017*). Combining the above findings, we propose that Bcas2 may have an evolutionarily conserved role in spermatogenesis.

Given the difficulty of obtaining homozygous mutants, embryos lacking one copy of *bcas2* gene were produced from crosses between heterozygous females and wild-type males. We observed a significant decrease of *bcas2* expression in the ICM region in *bcas2*$^{+/\Delta 7}$ or *bcas2*$^{+/\Delta 14}$ mutants, likely resulting from nonsense-mediated RNA decay (*Figure 1D*). In line with a previous report (*Yu et al., 2019*), a marked reduction in the expression of the HSC marker *cmyb* and T-cell marker *rag1* was found in *bcas2*$^{+/\Delta 7}$ or *bcas2*$^{+/\Delta 14}$ embryos at 5 dpf, indicating an essential role of *bcas2* in definitive hematopoiesis (*Figure 1—figure supplement 3A and B*). These findings suggest that *bcas2*$^{+/\Delta 7}$ and *bcas2*$^{+/\Delta 14}$ mutants can be used to examine the involvement of *bcas2* in primitive hematopoiesis.

To explore whether *bcas2* is required for primitive hematopoiesis, we first examined the expression of primitive erythropoietic markers *gata1* and *hbbe3* in *bcas2*$^{+/\Delta 7}$ and *bcas2*$^{+/\Delta 14}$ embryos at 22 hpf, and observed a marked decrease in the expression of these genes in the mutants (*Figure 1E and F*). Surprisingly, *o*-dianisidine staining showed similar hemoglobin contents in the *bcas2*$^{+/\Delta 7}$ and *bcas2*$^{+/\Delta 14}$ embryos at 48 hpf compared with control embryos, suggesting that the defect in primitive hematopoiesis induced by haploinsufficiency of *bcas2* was alleviated at later developmental stages. In order to further explore the role of BCAS2 in primitive hematopoiesis, we identified several *bcas2*$^{\Delta 14/\Delta 14}$ mutants from about 100 embryos. These homozygous mutants display a severe decrease in hemoglobin (*Figure 1G*). Moreover, injection of a translation-blocking MO into wild-type embryos to downregulate *bcas2* expression resulted in severe defects in erythropoiesis at 22 hpf and 48 hpf (*Figure 1—figure supplement 4A–D*). These results indicate that *bcas2* is indispensable for primitive hematopoiesis in zebrafish. In addition, transgenic mice expressing Cre recombinase under the control of the *Kdr* promoter were crossed to *Bcas2*$^{F/F}$ animals to induce the deletion of *Bcas2* in endothelial/ hematopoietic cells. We found that red blood cells were eliminated in the yolk sac of *Bcas2*$^{F/F}$;*Kdr-Cre* mice at E12.5 despite the presence of vessels (*Figure 1H*). Therefore, Bcas2 has a conserved role in vertebrates to regulate primitive hematopoiesis.

## *bcas2* deficiency impairs hematopoietic progenitor differentiation

The decrease in primitive hematopoietic cells in *bcas2* deficient animals may be attributed to a number of possible causes: excessive apoptosis, hampered proliferation of hematopoietic cells, or impaired differentiation of hematopoietic progenitor cells. To shed light on this issue, we first performed terminal deoxynucleotidyl transferase-mediated dUTP nick end labeling (TUNEL) assay

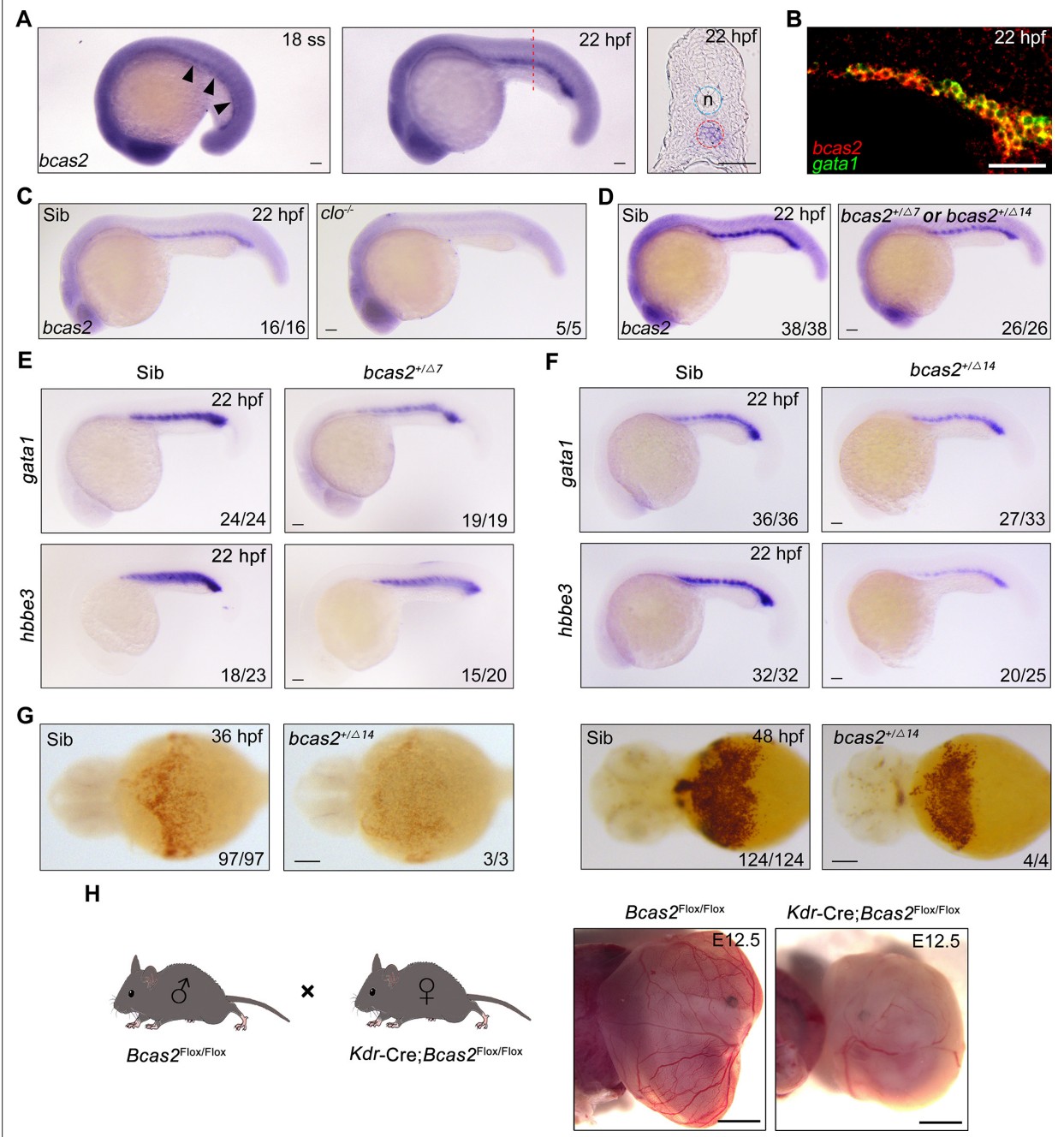

**Figure 1.** *bcas2* is expressed in the intermediate cell mass (ICM) and required for primitive hematopoiesis. (**A**) Whole-mount in situ hybridization (WISH) assay showing *bcas2* expression in the ICM at the 18-somite stage and 22 hpf. The dotted lines represent the section position and the black arrowheads indicate the ICM region. n, notochord. (**B**) Double fluorescence in situ hybridization (FISH) assay showing the expression pattern of *bcas2* and *gata1* in the ICM at 22 hpf. (**C, D**) Comparison of *bcas2* expression in *cloche* mutants (**C**) or *bcas2* heterozygous mutants (**D**) along with their corresponding siblings. (**E, F**) Expression analysis of *gata1* and *hbbe3* in *bcas2⁺/Δ7* and *bcas2⁺/Δ14* embryos. (**G**) Hemoglobin detection using *o*-dianisidine staining in *bcas2* homozygous mutant at 36 and 48 hpf. (**H**) Representative images of yolk sac from the hemangioblast-specific *Bcas2* knockout mice and their siblings. *Bcas2^F/F* females were crossed with *Bcas2^F/+;Kdr-Cre* males to induce the deletion of *Bcas2* in hemangioblasts. Scale bars, 100 µm (**A**, **C–G**), 50 µm (**B**), 1 mm (**H**).

The online version of this article includes the following source data and figure supplement(s) for figure 1:

**Figure supplement 1.** Expression patterns of *bcas2* in wild-type embryos during development.

**Figure supplement 2.** Zebrafish *bcas2* mutants are generated by using CRISPR/Cas9 system.

**Figure supplement 2—source data 1.** PDF file containing original gel images for *Figure 1—figure supplement 2C* with the relevant bands and

*Figure 1 continued on next page*

*Figure 1 continued*

treatments indicated.

**Figure supplement 2—source data 2.** Original gel images in *Figure 1—figure supplement 2C*.

**Figure supplement 3.** *bcas2* is essential for definitive hematopoiesis.

**Figure supplement 4.** Knockdown of *bcas2* impairs primitive hematopoiesis.

**Figure supplement 4—source data 1.** Original western blots for *Figure 1—figure supplement 4A* with the relevant bands and treatments indicated.

**Figure supplement 4—source data 2.** Original western blot images in *Figure 1—figure supplement 4A*.

in *Tg(gata1:GFP)* embryos at the 10-somite stage to examine DNA fragmentations in apoptotic cells and found no obvious apoptotic signal in the *gata1*+ cells in either *bcas2*+/Δ14 embryos or their wild-type siblings (*Figure 2—figure supplement 1A*). Meanwhile, BrdU incorporation assay revealed no significant difference in hematopoietic cell proliferation between *bcas2*+/Δ14 mutants and their corresponding wild-types (*Figure 2—figure supplement 1B and C*). These data suggest that *bcas2* is dispensable for the survival and proliferation of hematopoietic cells.

In the developing embryo, hemangioblasts are derived from the ventral mesoderm at early somite stage and then differentiate into both hematopoietic and endothelial lineages (*Vogeli et al., 2006*; *Reischauer et al., 2016*). To test whether *bcas2* functions in cell fate decision during primitive

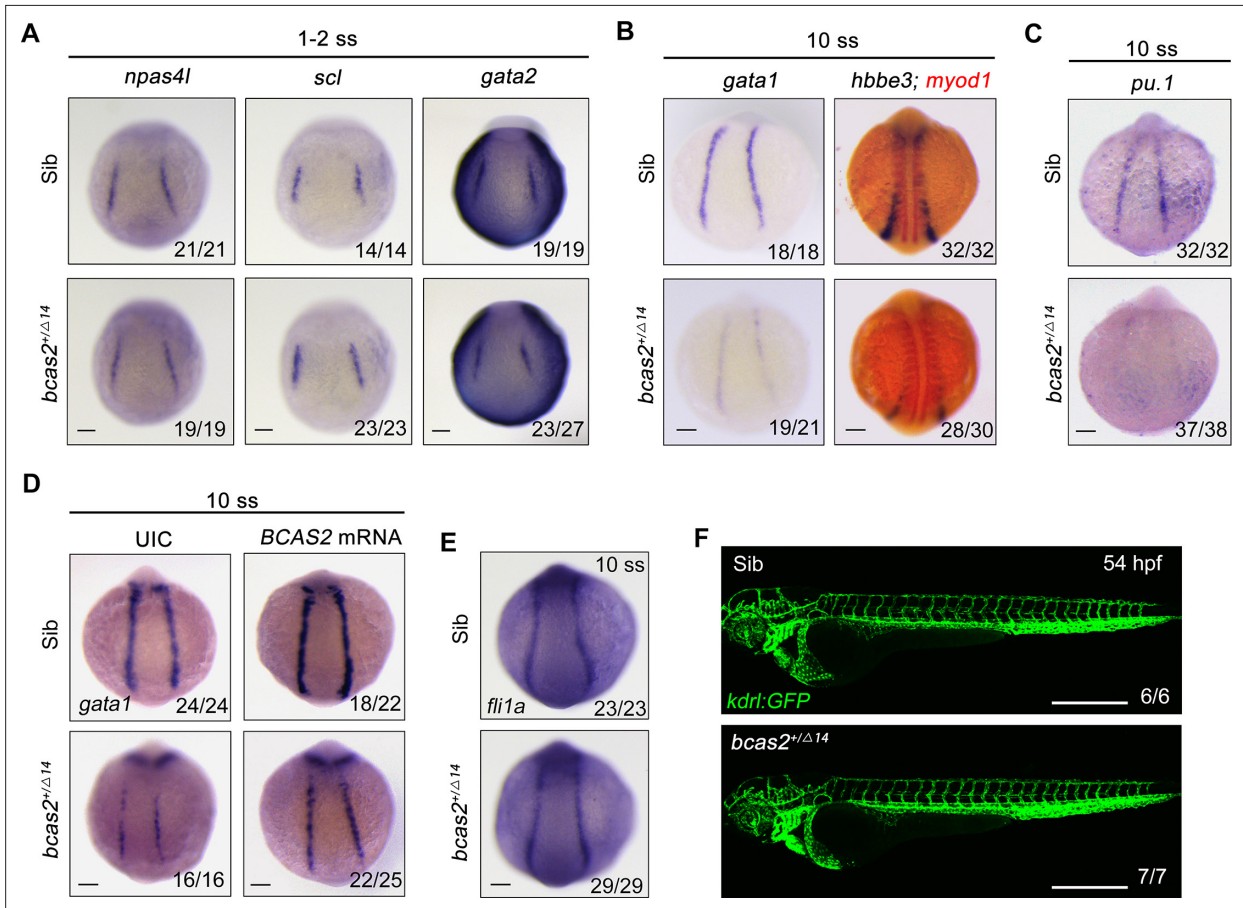

**Figure 2.** *bcas2* is required for hematopoietic progenitor differentiation. (**A–C**) Expression analysis of hemangioblast markers *npas4l*, *scl*, *gata2* (**A**), erythroid progenitor markers *gata1*, *hbbe3* (**B**), and myeloid marker *pu.1* (**C**) in *bcas2*+/Δ14 embryos and their wild-type siblings at indicated stages. (**D**) Expression changes of *gata1* in *bcas2*+/Δ14 embryos overexpressing BCAS2 at the 10-somite stage. The indicated embryos were injected with or without 300 pg of human *BCAS2* mRNA at the one-cell stage. (**E**) Expression of endothelial marker *fli1a* in *bcas2*+/Δ14 and sibling embryos at the 10-somite stage. (**F**) Confocal imaging of *bcas2*+/Δ14 and control sibling *Tg(kdrl:GFP)* embryos at 54 hpf. Scale bars, 100 μm (**A–E**), 500 μm (**F**).

The online version of this article includes the following figure supplement(s) for figure 2:

**Figure supplement 1.** *bcas2* is dispensable for the survival and proliferation of hematopoietic cells.

hematopoietic cell development, the expression of hemangioblast markers *npas4l*, *scl*, and *gata2* in *bcas2*[+/Δ14] embryos was examined at the 1- to 2-somite stage. As shown in *Figure 2A*, haploinsufficiency of *bcas2* did not affect the emergence of the hemangioblast population. Then we extended our analysis to include the markers of hematopoietic and endothelial progenitors. Consistent with the decrease in primitive hematopoietic cells in *bcas2* deficient mutants, a marked reduced expression of erythrocyte progenitor markers *gata1* and *hbbe3* was observed in the PLM of *bcas2*[+/Δ14] embryos at the 10-somite stage (*Figure 2B*). Interestingly, the expression of myeloid progenitor marker *pu.1* was also dramatically decreased (*Figure 2C*). Moreover, overexpression of human BCAS2 enhanced the expression of *gata1* in both wild-type and mutant embryos at the 10-somite stage (*Figure 2D*). In contrast, the endothelial progenitor marker *fli1a* was expressed at a similar level in *bcas2*[+/Δ14] embryos as in wild-type animals (*Figure 2E*). Consistently, blood vessels appeared normal in *bcas2*[+/Δ14] mutants with *Tg(kdrl:GFP)* background at 54 hpf (*Figure 2F*). These data provide convincing evidence that *bcas2* is required for the differentiation of the hematopoietic lineage from hemangioblasts during primitive hematopoiesis.

## BCAS2 functions in primitive hematopoiesis by activating Wnt signaling

Previous studies have shown that Wnt/β-catenin plays a key role in primitive hematopoiesis (*Tran et al., 2010*; *Lengerke et al., 2008*; *Sun et al., 2018*). As both BCAS2 and β-catenin-like 1 (CTNNBL1) are members of the Prp19/CDC5L complex, which is a major building block of the spliceosome's catalytic RNP core (*Grote et al., 2010*), we speculate that BCAS2 may be a regulator of Wnt signaling through interaction with β-catenin during hematopoiesis. To test our hypothesis, human BCAS2 was overexpressed in HEK293T cells and mouse embryonic fibroblasts (MEFs). Ectopic expression of BCAS2 enhanced the Wnt3a-induced expression of the TOPflash luciferase reporter in a dose-dependent manner (*Figure 3A and B*). Importantly, Wnt3a-induced luciferase activity in HEK293T cells could be effectively reduced by knockdown of *BCAS2* using two shRNAs targeting different regions of human *BCAS2* (*Figure 3C*, *Figure 3—figure supplement 1*). Similar results were also observed in conditional *Bcas2* knockout (*Bcas2*-cKO) MEFs in the presence of tamoxifen (*Figure 3D*). Furthermore, the expression of *cdx4* and *hoxa9a*, which are targets of canonical Wnt signaling in the regulation of hematopoietic development (*Pilon et al., 2006*, *Sun et al., 2018*), were downregulated in the lateral plate mesoderm of *bcas2*[+/-] embryos at the six-somite stage (*Figure 3—figure supplement 2*). These findings support that BCAS2 promotes Wnt signaling activation.

To confirm that Wnt signaling was required for zebrafish embryonic hematopoiesis, we induced the expression of canonical Wnt inhibitor Dkk1 by heat-shocking *Tg(hsp70l:dkk1-GFP)*[w32] embryos at the bud stage (*Stoick-Cooper et al., 2007*). As expected, diminished expression of *gata1* and *hbbe3* was detected in the resulting embryos at the 10-somite stage (*Figure 3E and F*). In addition, treatment with a small molecule β-catenin antagonist CCT036477 from 9 hpf did not affect the expression of hemangioblast markers *npas4l*, (*Krause et al., 2014*) *scl*, and *gata2* or endothelial marker *fli1a* (*Figure 3—figure supplement 3A and B*), yet significantly reduced the expression of erythroid progenitor marker *gata1* in wild-type embryos (*Figure 3—figure supplement 3C*), suggesting that canonical Wnt signaling may not be required for the generation of hemangioblasts or their endothelial differentiation, but is pivotal for their hematopoietic differentiation. To further validate that *bcas2* functions in primitive hematopoiesis via Wnt/β-catenin signaling, the expression pattern of β-catenin was examined in *bcas2* morphants with *Tg(gata1:GFP)* background at the 10-somite stage by immunofluorescence staining. The signal of nuclear β-catenin was substantially decreased in hematopoietic progenitor cells (*Figure 3G and H*) and primitive myeloid cells (*Figure 3—figure supplement 4A and B*). Moreover, overexpression of ΔN-β-catenin, a constitutively active form of β-catenin, effectively restored the expression of *hbbe3* in *bcas2* morphants and mutants (*Figure 3I and J*). All these data suggest that BCAS2 functions in primitive hematopoiesis by regulating Wnt/β-catenin signaling.

## BCAS2 promotes β-catenin nuclear accumulation independently of protein stability regulation

To investigate how BCAS2 regulates Wnt/β-catenin signaling, HEK293T cells were treated with LiCl, a canonical Wnt agonist that inhibits GSK-3β activity and stabilizes cytosolic β-catenin (*Meijer et al., 2003*). The results showed that TOPflash activity was significantly elevated in LiCl-treated cells (*Figure 4A*). BCAS2 overexpression further upregulated, whereas shRNA-mediated knockdown of

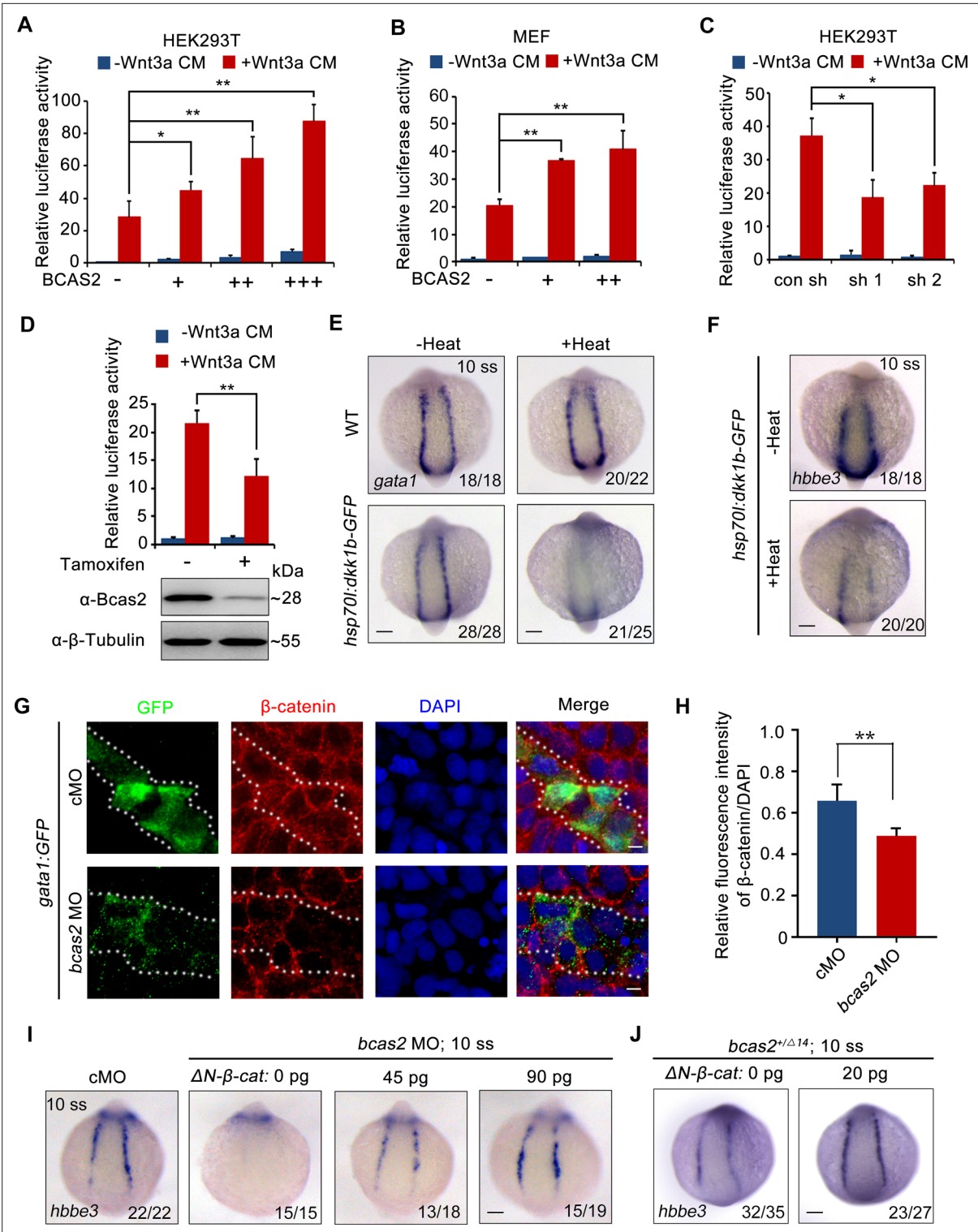

**Figure 3.** BCAS2 promotes primitive hematopoiesis via activating Wnt signaling. (**A, B**) Overexpression of BCAS2 increases Wnt3a-induced TOPflash activity in HEK293T cells (**A**) and mouse embryonic fibroblasts (MEFs) (**B**). Different amounts of plasmid expressing BCAS2 (0, 80, 160, or 320 ng/well) were transfected into cells, together with the super-TOPflash luciferase and Renilla luciferase vectors. After 36 h of transfection, cells were treated with or without Wnt3a CM for 12 h and harvested for luciferase assays (n=3). *p<0.05; **p<0.01 (Student's *t*-test). (**C**) The Wnt3a-induced TOPflash activity is decreased in BCAS2-deficient cells. HEK293T cells were transfected with shRNA plasmids, along with indicated plasmids, and harvested for luciferase reporter assay (n=3). *p<0.05 (Student's *t*-test) (**D**) *Bcas2*-cKO MEFs prepared from *Bcas2^F/F^* mouse embryos were incubated in medium containing 100 μM tamoxifen for 72 h and then subjected to western blotting and luciferase reporter assay (n=3). **p<0.01 (Student's *t*-test). (**E, F**) Expression

*Figure 3 continued on next page*

*Figure 3 continued*

analysis of *gata1* (**E**) and *hbbe3* (**F**) in *Tg(hsp70l:dkk1b-GFP)* embryos after heat shock at 16 hpf. (**G, H**) Immunofluorescence staining of β-catenin in *Tg(gata1:GFP)* embryos at 16 hpf. The embryos were injected with 8 ng of the indicated MO at the one-cell stage. The dotted lines show the GFP-positive hematopoietic progenitor cells. The relative fluorescence intensity of nuclear β-catenin was quantified in (**H**) (n=6). **p<0.01 (Student's *t*-test). (**I, J**) Expression of *hbbe3* in *bcas2* morphants (**I**) and *bcas2⁺/ᐃ¹⁴* mutants (**J**) overexpressing ΔN-β-catenin. Embryos were injected with the indicated MO together with *ΔN-β-catenin* mRNA at the 1-cell stage and harvested at the 10-somite stage for in situ hybridization. Scale bars, 100 μm (**E, F, I, J**), 5 μm (**G**).

The online version of this article includes the following source data and figure supplement(s) for figure 3:

**Source data 1.** Original western blots for *Figure 3D*, indicating the relevant bands and treatments.

**Source data 2.** Original western blot images in *Figure 3D*.

**Figure supplement 1.** Western blot analysis of HEK293T cells transfected with corresponding shRNA constructs.

**Figure supplement 1—source data 1.** Original western blots for *Figure 3—figure supplement 1* with the relevant bands labeled.

**Figure supplement 1—source data 2.** Uncropped immunoblotting images in *Figure 3—figure supplement 1*.

**Figure supplement 2.** Expression patterns of *cdx4* and *hoxa9a* in *bcas2⁺/ᐃ¹⁴* embryos and their siblings at the 6-somite stage.

**Figure supplement 3.** Inhibition of Wnt signaling does not affect the generation of hemangioblasts or their endothelial differentiation, but impairs their hematopoietic differentiation.

**Figure supplement 4.** Knockdown of *bcas2* significantly reduces nuclear β-catenin in the primitive myeloid cells.

*BCAS2* downregulated LiCl-induced TOPflash activity (*Figure 4A and B*). Likewise, HEK293T cells transfected with S37A-β-catenin, a constitutively active form of β-catenin that is resistant to GSK-3β-mediated degradation (*Easwaran et al., 1999*), displayed a much higher level of TOPflash activity, which was reduced by *BCAS2* knockdown (*Figure 4C*). These results strongly imply that BCAS2 regulates Wnt signaling downstream of β-catenin stability control.

To test the above hypothesis, we evaluated nuclear β-catenin level by performing immunofluorescence staining and immunoblotting experiments. Upon tamoxifen exposure, nuclear accumulation of β-catenin induced by LiCl was greatly inhibited in *Bcas2*-cKO MEFs, while nuclear/cytoplasmic fractionation suggested that cytoplasmic β-catenin level remained relatively unchanged (*Figure 4D and E*). Similarly, silencing *BCAS2* with shRNA led to reduced nuclear β-catenin in the human colon cancer cell line SW480, in which β-catenin was activated because of mutations in the adenomatous polyposis coli protein (APC), an integral component of the β-catenin destruction complex (*Figure 4F*, *Rosin-Arbesfeld et al., 2003*). Next, MG132, a proteasome inhibitor, was applied to activate Wnt/β-catenin signaling in *Bcas2*-cKO MEFs by inhibiting β-catenin degradation. In the absence of tamoxifen and MG132, endogenous β-catenin was localized almost exclusively in the cytoplasm; MG132 treatment dramatically triggered β-catenin accumulation in the nuclei (*Figure 4G*). However, in *Bcas2*-cKO MEFs exposed to tamoxifen, MG132 treatment was not able to induce nuclear accumulation of β-catenin (*Figure 4G*). These findings indicate that BCAS2 promotes β-catenin nuclear accumulation in a manner that is independent of β-catenin stability regulation.

## BCAS2 sequesters β-catenin within the nucleus

In addition to be affected by protein stability, the nuclear level of β-catenin is also fine-tuned by the opposing actions of nuclear export and import (*Lu et al., 2017*; *Henderson and Fagotto, 2002*; *Henderson, 2000*). To examine the effect of BCAS2 on the nuclear import and export of β-catenin, fluorescent recovery after photobleaching (FRAP) experiments were carried out in HeLa cells expressing GFP-tagged S37A-β-catenin. After photobleaching the nucleus, no significant difference was found in the recovery of nuclear GFP signals between the cells with and without overexpression of BCAS2 (*Figure 5—figure supplement 1A, A' and C*), suggesting that BCAS2 does not regulate β-catenin nuclear import. Conversely, after photobleaching the cytoplasm, BCAS2 overexpressed cells showed a much slower recovery of cytoplasmic fluorescence (*Figure 5—figure supplement 1B, B', and C*), indicating that BCAS2 inhibits β-catenin nuclear export.

It has been suggested that the nuclear exit of β-catenin can be either dependent or independent on CRM1, a major nuclear export receptor (*Xu and Massagué, 2004*). To shed light on the mechanism underlying BCAS2 mediated β-catenin nuclear retention, we treated *Bcas2*-cKO MEFs with the CRM1-specific export inhibitor leptomycin B (LMB) (*Wolff et al., 1997*). Regardless of the presence or absence of endogenous BCAS2, LMB treatment could effectively increase the level of β-catenin in

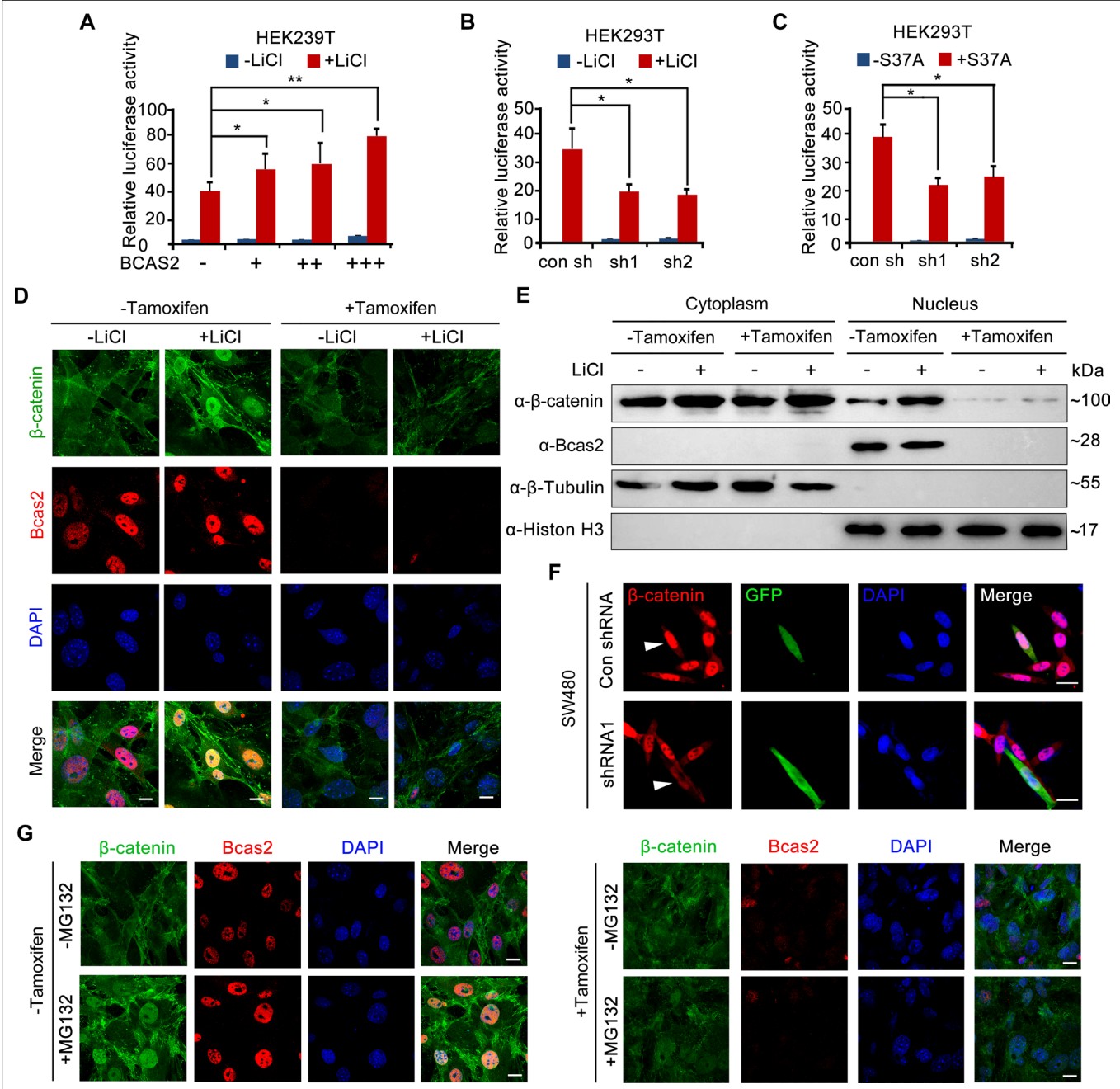

**Figure 4.** BCAS2 is essential for β-catenin nuclear accumulation. (A–C) BCAS2 enhances LiCl-induced TOPflash activity in HEK293T cells. Cells were transfected with BCAS2 expression plasmids (**A**), shRNA plasmids (**B**), or S37A-β-catenin expression plasmids (**C**), together with the TOPflash luciferase and Renilla luciferase vectors. After transfection, cells were subsequently treated with or without 100 ng/ml LiCl for 12 h and assayed for luciferase activity (n=3). *p<0.05; **p<0.01 (Student's *t*-test). (**D, E**) *Bcas2*-cKO mouse embryonic fibroblasts (MEFs) were incubated with tamoxifen for 24 h and then treated with or without 100 ng/mL LiCl. The nuclear accumulation of β-catenin was analyzed using immunofluorescence (**D**) and western blotting (**E**). (**F**) SW480 cells were transfected with the indicated shRNA constructs, and the endogenous β-catenin protein was detected using immunofluorescence 48 h after transfection. The expression of GFP served as a transfection control. The arrowheads indicate the cells transfected with indicated shRNA constructs. (**G**) *Bcas2*-cKO MEFs were cultured in the presence of tamoxifen for 24 h and then treated with 20 µM MG132 for 6 h. The expression of BCAS2 and β-catenin was measured by immunofluorescence. Scale bars, 10 µm (**D, F, G**).

The online version of this article includes the following source data for figure 4:

**Source data 1.** Original western blots for *Figure 4E* with the relevant bands and treatments labeled.

**Source data 2.** Original western blot images in *Figure 4E*.

the nucleus (*Figure 5A*). Consistently, treatment of LMB was able to rescue the impaired nuclear accumulation of β-catenin in BCAS2-deficient SW480 cells (*Figure 5B*). Moreover, when *bcas2* morphants in *Tg (gata1:GFP)* background were treated with LMB from bud stage to 10 somite stage, the level of nuclear β-catenin was partially recovered (*Figure 5C and D*). Importantly, the expression of *gata1* was also restored in *bcas2* mutants upon LMB treatment (*Figure 5E*). We further tested if BCAS2 specifically regulates CRM1-mediated nuclear export of β-catenin by analyzing the nucleocytoplasmic distribution of other known CRM1 cargoes, such as ATG3 and CDC37L (*Kırlı et al., 2015*). Intriguingly, BCAS2 overexpression in HeLa cells only slightly enhanced the nuclear localization of CDC37L and had no significant impact on that of ATG3 (*Figure 5—figure supplement 2*), indicating the specificity of BCAS2-mediated inhibition of CRM1-dependent nuclear export of β-catenin. Taken together, these findings suggest that BCAS2 negatively regulates CRM1-mediated nuclear export of β-catenin.

## BCAS2 directly interacts with β-catenin in the nucleus

To investigate whether BCAS2 inhibits the nuclear export of β-catenin through physical binding, HEK293T cells were transfected with Flag-tagged β-catenin and HA-tagged BCAS2 constructs. Co-immunoprecipitation (Co-IP) experiments showed that Flag-β-catenin was precipitated with HA-BCAS2 as well as endogenous BCAS2, indicating an interaction between these two proteins (*Figure 6A and B*). In addition, the interaction was enhanced upon Wnt3a stimulation (*Figure 6C*). Given that Wnt ligand stimulation ultimately induces β-catenin nuclear accumulation, this enhanced interaction implies that BCAS2 associates with β-catenin within the nucleus. Therefore, we performed the bimolecular fluorescence complementation (BiFC) assay to visualize the interaction of BCAS2 and β-catenin in living cells. In this assay, the N-terminal fragment of yellow fluorescent protein (YFP) was fused to BCAS2 (YN-BCAS2), while the C-terminal fragment was fused to β-catenin (YC-β-catenin) (*Figure 6D*). As expected, the YFP fluorescence was specifically observed in the nucleus (*Figure 6E*).

Previous studies have divided the β-catenin protein into three distinct domains, including the N-terminal domain (residues 1–133), the central domain with 12 ARM repeats (residues 134–670), and the C-terminal domain (residues 671–781) (*Dimitrova et al., 2010*). To identify the BCAS2 binding site, constructs expressing various truncated forms of β-catenin were generated and co-transfected with BCAS2 into HEK293T cells (*Figure 6F*). Co-IP assays revealed that deletion of the N-terminal or C-terminal domain of β-catenin did not alter the interaction between β-catenin and BCAS2 (*Figure 6G*). In contrast, when the ARM repeats 1–12 of β-catenin were deleted, the resulting deletion mutant showed virtually no interaction with BCAS2 (*Figure 6G*). GST pull-down assay also demonstrated a direct interaction between BCAS2 and the ARM repeats of β-catenin (*Figure 6H*). These results indicate that BCAS2 physically binds to the ARM repeats of β-catenin. Furthermore, we found that the ARM repeats 9–12, but not 1–8, bound to BCAS2 (*Figure 6G*).

Transcriptional activation of the canonical Wnt target genes depends on β-catenin nuclear localization and its physical association with TCF/LEF family members. As the binding sites for TCF have been located in the ARM repeats 3–10 of β-catenin, (*Graham et al., 2000*) it is likely that BCAS2-mediated nuclear sequestration of β-catenin through interacting with the ARM repeats 9–12 would be compatible with the initiation of gene transcription by allowing for the association of β-catenin and TCF. To validate this possibility, co-IP assays were performed and we found that β-catenin still bound with TCF4 in the presence of BCAS2 (*Figure 6—figure supplement 1*), confirming that the binding of BCAS2 to β-catenin would not interfere with the formation of β-catenin/TCF complex.

## BCAS2 enhances β-catenin nuclear accumulation through its CC domains

To determine which domain of BCAS2 binds to β-catenin, we constructed a series of deletion mutants of BCAS2 (*Figure 7A*). Notably, we observed that among these truncated mutants, only the one lacking both CC1 and CC2 domains lost the ability to interact with β-catenin (*Figure 7B*). Moreover, these two CC domains alone or together could interact with β-catenin (*Figure 7C*). Therefore, we conclude that BCAS2 binds to β-catenin via its CC domains.

We next examined whether the CC domains are required for BCAS2 to promote Wnt/β-catenin signaling. As shown in *Figure 7D*, overexpression of BCAS2 without the CC domains failed to increase LiCl-induced TOPflash activity in HEK-293T cells. Likewise, overexpression of the full-length or the CC domains alone, but not BCAS2 lacking the CC domains, restored the nuclear accumulation of

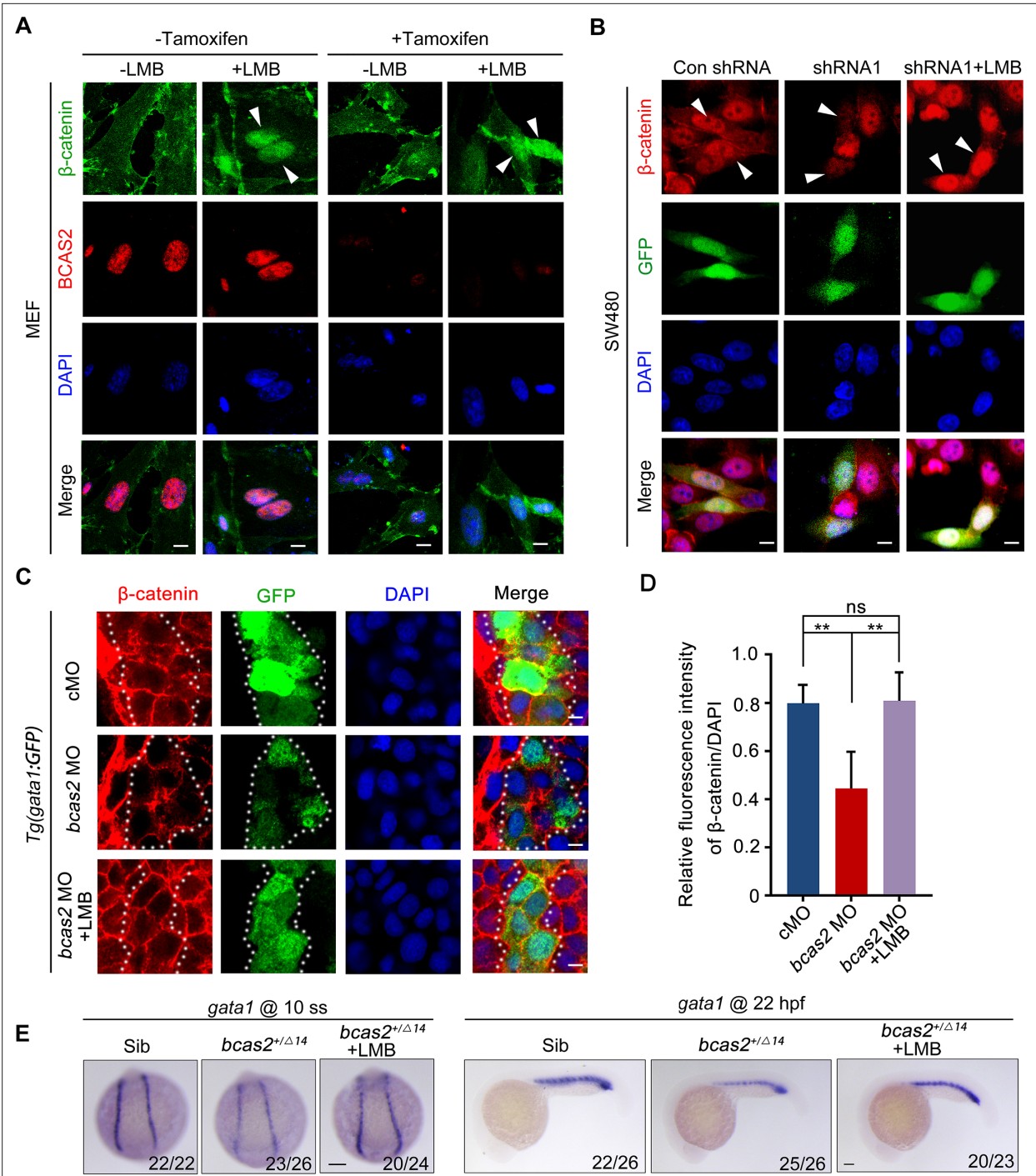

**Figure 5.** BCAS2 functions in CRM1-mediated nuclear export of β-catenin. (**A**) Tamoxifen-treated *Bcas2*-cKO mouse embryonic fibroblasts (MEFs) were incubated with 20 nM LMB for 3 h. The expression of Bcas2 and β-catenin was analyzed using immunofluorescence. The arrowheads show the cells with nuclear β-catenin accumulation. (**B**) SW480 cells were transfected with the indicated shRNA constructs and then treated with LMB for 3 h before immunostaining. GFP was regarded as a transfection control. The arrowheads indicate the transfected cells. (**C, D**) Immunofluorescence staining of β-catenin in *bcas2* morphants with *Tg(gata1:GFP)* background at 16 hpf. Embryos were exposed to 20 nM LMB from the bud stage. The dotted lines indicate the GFP-positive hematopoietic progenitor cells. The relative fluorescence intensity of nuclear β-catenin was quantified in (**D**) (n=6). ns, not significant; \*\*p<0.01 (Student's *t*-test). (**E**) *bcas2*^+/Δ14^ embryos were treated with 20 nM LMB for 6 h and then subjected to WISH assay to analyze the expression of *gata1* at the indicated stages. Scale bars, 10 μm (**A, B**), 5 μm (**C**), 100 μm (**E**).

The online version of this article includes the following figure supplement(s) for figure 5:

**Figure supplement 1.** BCAS2 inhibits the nuclear export of β-catenin.

*Figure 5 continued on next page*

*Figure 5 continued*

**Figure supplement 2.** Overexpression of BCAS2 slightly enhances the nuclear accumulation of CDC37L and has no influence on the distribution of ATG3.

β-catenin in *bcas2* morphants (*Figure 7E-F*, *Figure 7—figure supplement 1A and B*). The expression of *gata1* in *bcas2* mutants was also recovered by overexpression of the full-length BCAS2, but not the truncated form without the CC-domains (*Figure 7G*). Collectively, these findings indicate that BCAS2 positively regulates Wnt signaling through sequestering β-catenin within the nucleus via its CC domains during primitive hematopoiesis.

As BCAS2 is involved in the Prp19-CDC5L spliceosome complex that regulates RNA splicing during spermiogenesis, neurogenesis, and definitive hematopoiesis, (*Liu et al., 2017*; *Yu et al., 2019*) we wondered if this protein participates in primitive hematopoiesis via mRNA alternative splicing. To this end, we performed RNA sequencing of 10-somite stage embryos to identify abnormal events in alternative splicing in $bcas2^{+/\Delta14}$ mutants. However, upon haploinsufficiency of *bcas2*, neither the number of five major types of alternative splicing events, nor the typical forms of alternative splicing were significantly affected (*Figure 7—figure supplement 2A and B*). Additionally, haploinsufficiency of *bcas2* did not result in the alternative splicing of *mdm4* that predisposes cells to undergo p53-mediated apoptosis in definitive hematopoiesis, as reported previously by Yu et al. (*Figure 7—figure supplement 2C*; *Yu et al., 2019*; *Rallapalli et al., 1999*). Furthermore, the splicing efficiency of β-catenin pre-mRNA remained almost unchanged in $bcas2^{+/\Delta14}$ mutants (*Figure 7—figure supplement 2D*). These results demonstrate that the defects in primitive hematopoiesis of $bcas2^{+/\Delta14}$ mutants are independent of the regulatory role of Bcas2 in pre-mRNA splicing.

## Discussion

BCAS2 is a 26 kDa nuclear protein involved in a multitude of developmental processes, such as *Drosophila* wing development, dendritic growth, and spermatogenesis (*Kuo et al., 2009*; *Chen et al., 2013*; *Liu et al., 2017*; *Huang et al., 2016*; *Xu et al., 2015*; *Zhang et al., 2022*). In our study, we generated *bcas2* knockout zebrafish. The heterozygotes not only showed male infertility, resembling the phenotype of *Bcas2* germ cell-specific knockout mice reported previously (*Liu et al., 2017*), but also exhibited impaired definitive hematopoiesis, consistent with the earlier study (*Yu et al., 2019*). Importantly, we found a marked decrease in the expression of the primitive erythroid progenitor markers *gata1* and *hbbe3* in these heterozygous mutants, which was rescued by overexpression of BCAS2. Moreover, the defective primitive hematopoiesis in mutant zebrafish was phenocopied in hemangioblast-specific *Bcas2* knockout mice. While the reason(s) for the discrepancy between our data and the observations made by Yu et al. regarding the role of *bcas2* in the development of primitive erythroid and myeloid cells remains to be determined (*Yu et al., 2019*), our findings in zebrafish and mouse embryos provide solid evidence that BCAS2 plays a conserved role in primitive hematopoiesis.

As demonstrated in previous studies, BCAS2 is involved in various developmental events by regulating pre-mRNA splicing (*Chen et al., 2013*; *Liu et al., 2017*; *Yu et al., 2019*; *Chen et al., 2022*; *Huang et al., 2016*). However, our data showed that haploinsufficiency of *bcas2* did not affect alternative splicing during primitive hematopoiesis. These results imply that one copy of the *bcas2* gene is sufficient to support mRNA splicing in zebrafish. Instead, we find that Bcas2 promotes primitive hematopoiesis by sequestering β-catenin within the nucleus. It has been reported that the *bcas2* deletion in zebrafish embryos induces alternative splicing of *Mdm4* that predisposes cells to undergo p53-mediated apoptosis in HSPCs during definitive hematopoiesis (*Yu et al., 2019*). Intriguingly, we found that the loss of one copy of *bcas2* gene in zebrafish also resulted in severe impairment of HSPCs and their derivatives. It is possible that Bcas2 might also have a role in definitive hematopoiesis independent of its splicing regulatory function.

For the past decades, given the contradictory conclusions obtained from various in vitro and in vivo studies, the function of Wnt/β-catenin signaling in primitive hematopoiesis remains elusive and controversial (*Sturgeon et al., 2014*; *Tran et al., 2010*; *Lengerke et al., 2008*; *Paluru et al., 2014*). In the present study, we have provided several lines of evidence supporting that Wnt/β-catenin signaling positively regulates primitive hematopoiesis: (1) inhibition of Wnt/β-catenin by overexpression of

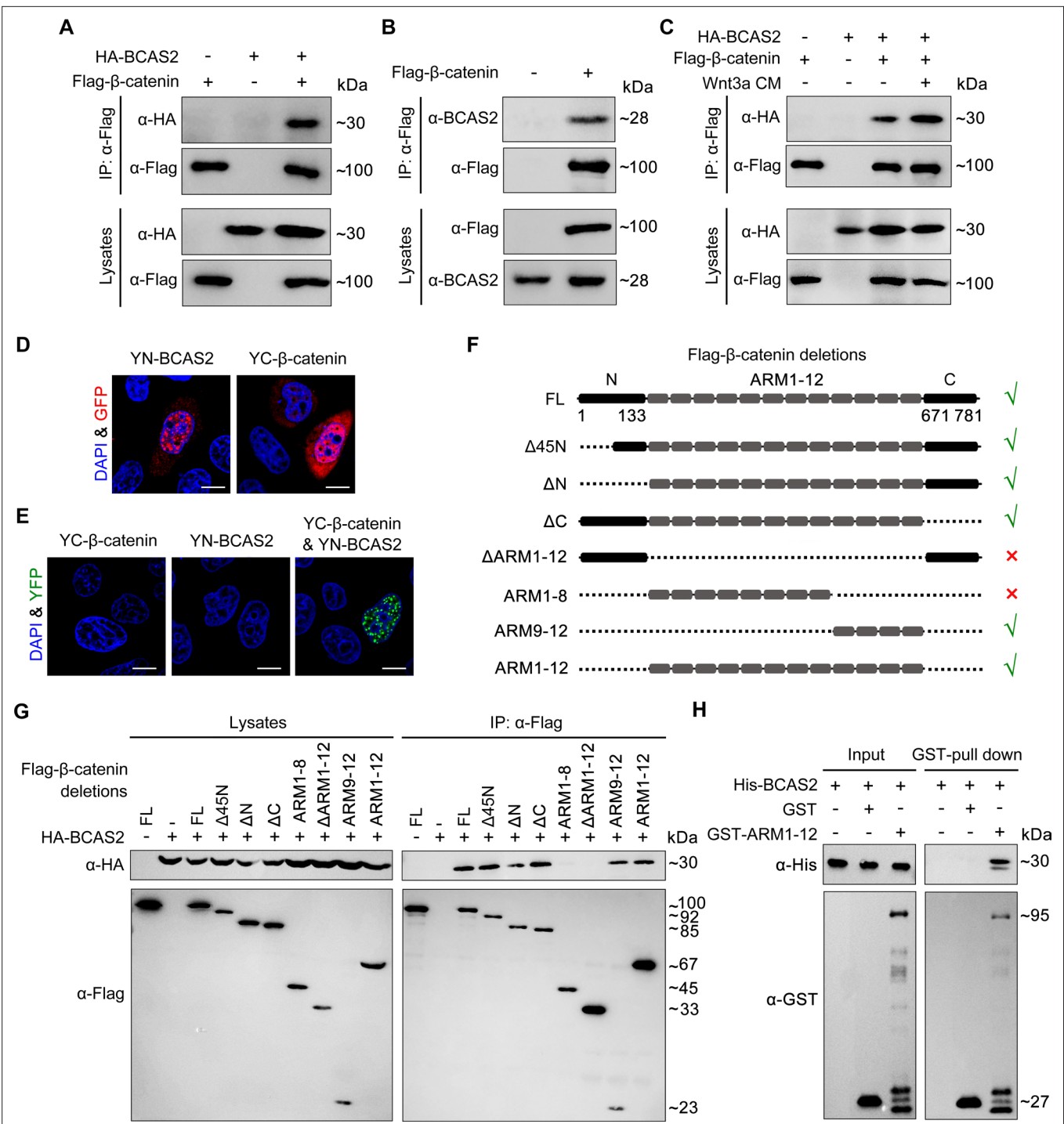

**Figure 6.** BCAS2 interacts with β-catenin. (**A–C**) Flag-tagged β-catenin was co-transfected with or without HA-tagged BCAS2 into HEK293T cells. Cell lysates were immunoprecipitated using anti-Flag antibody. Eluted proteins were analyzed by western blotting using indicated antibodies. In (**C**), for Wnt signaling activation, cells were treated with Wnt3a CM for 5 h before harvest. (**D, E**) YN-BCAS2 and YC-β-catenin were either individually or collectively transfected into HeLa cells. The expression of YN-BCAS2 and YC-β-catenin was analyzed with anti-GFP antibody (**D**). The reconstituted YFP fluorescence in living cells was detected by confocal laser scanning microscopy with excitation at 488 nm (**E**). (**F**) Schematics of full-length and deletion mutants of β-catenin. (**G**) HEK293T cells were transfected with HA-tagged BCAS2 and Flag-tagged deletion mutants of β-catenin. Cell lysates were then immunoprecipitated using anti-Flag antibody followed by western blot analysis. (**H**) GST pull-down assays were performed using bacterially expressed GST, GST-ARM1-12, and His-BCAS2. Scale bars, 10 μm (**D, E**).

The online version of this article includes the following source data and figure supplement(s) for figure 6:

*Figure 6 continued on next page*

*Figure 6 continued*

**Source data 1.** Original western blots for *Figure 6A–C, G and H* with the relevant bands and treatments indicated.

**Source data 2.** Uncropped immunoblotting images in *Figure 6A–C, G and H*.

**Figure supplement 1.** The interaction between β-catenin and TCF4 remains unaffected in the presence of BCAS2.

**Figure supplement 1—source data 1.** Original western blots for *Figure 6—figure supplement 1*, indicating the relevant bands and treatments.

**Figure supplement 1—source data 2.** Original files for western blot analysis in *Figure 6—figure supplement 1*.

the canonical Wnt inhibitor Dkk1 disrupts the formation of erythrocyte progenitors at the 10-somite stage. (2) Defects in primitive hematopoiesis in *bcas2* morphants and mutants are readily restored by overexpression of ΔN-β-catenin, a constitutively active β-catenin. (3) Overexpression of the full-length BCAS2, but not the CC domain-deleted BCAS2, restores the formation of the primitive erythroid progenitor in *bcas2* mutants. (4) BCAS2 overexpression enhances the development of primitive blood cells in wild-type embryos. All these data suggest that BCAS2-mediated Wnt/β-catenin signal activation is necessary for primitive hematopoiesis.

In addition, Wnt/β-catenin signaling has been known as an important pathway involved in the regulation of axis determination and neural patterning during gastrulation (*Yamaguchi, 2001*; *Kozmikova and Kozmik, 2020*; *Brafman and Willert, 2017*; *Lickert et al., 2005*). In our study, neither the heterozygous *bcas2* mutant embryos nor the very few homozygous ones exhibited any morphological defects typically associated with inhibition of Wnt signaling, such as ventralization or brain anteriorization. This may be due to the presence of maternal Bcas2 in heterozygous and homozygous mutant embryos which were derived from crossing *bcas2* heterozygous adult zebrafish.

CRM1 can facilitate β-catenin nuclear export in distinct ways (*Morgan et al., 2014*). For example, CRM1 usually recognizes and binds with the nuclear export signal (NES) sequences in chaperon proteins, such as APC, Axin, and Chibby (*Neufeld et al., 2000*; *Cong and Varmus, 2004*; *Li et al., 2008*), to mediate the nuclear export of β-catenin. On the other hand, CRM1 can also bind directly to and function as an efficient nuclear exporter for β-catenin (*Ki et al., 2008*). Since BCAS2 has not been reported to contain any recognizable NES sequences, it will be interesting to investigate whether BCAS2 competitively inhibits β-catenin from associating with CRM1, or with the chaperone proteins.

In summary, we uncover a novel role of BCAS2 in primitive hematopoiesis through enhancing nuclear retention of β-catenin. Our study provides new insights into the mechanism of BCAS2-mediated Wnt signal activation during primitive hematopoiesis. Given that BCAS2 and Wnt signaling are well documented to contribute to cancer development (*Murillo-Garzón and Kypta, 2017*; *Zhan et al., 2017*; *Yu et al., 2021*; *Salmerón-Hernández et al., 2019*; *Wang et al., 2020*), it is appealing to further explore whether our findings can be applied to future cancer research.

## Materials and methods

### Animal models

Our studies, including animal maintenance and experiments, were performed in compliance with the guidelines of the Animal Care and Use Committee of the South China University of Technology (Permission Number: 2023092). Seven strains of zebrafish were used in this study, including Tübingen wild-type, *bcas2* mutant, *cloche* mutant, *Tg(gata1:GFP)*, *Tg(coro1a:eGFP)*, *Tg(kdrl:GFP)*, and *Tg(hsp70l:dkk1b-GFP)*. *cloche* mutant, *Tg(gata1:GFP)* and *Tg(kdrl:GFP)* lines were provided by Professor Feng Liu (Chinese Academy of Sciences). *Tg(coro1a:eGFP)* was provided by Professor Yiyue Zhang (South China University of Technology). *Tg(hsp70l:dkk1b-GFP)* strain was purchased from the China Zebrafish Resource Center. *Bcas2*^Floxed/Floxed^ (*Bcas2*^F/F^) mouse line was generated as previously described (*Liu et al., 2017*). *Kdr*-Cre mouse line was provided by Professor Dahua Chen (Yunnan University). Genotyping of *Bcas2*^F/F^ mouse and *Kdr*-Cre mouse was performed using primers listed in *Supplementary file 1*. The mouse model with *Bcas2* specifically disrupted in the hemangioblasts was derived from mating female *Bcas2*^F/F^ mice with *Kdr*-Cre transgenic mice. All mouse lines were maintained on a mixed background (129/C57BL/6).

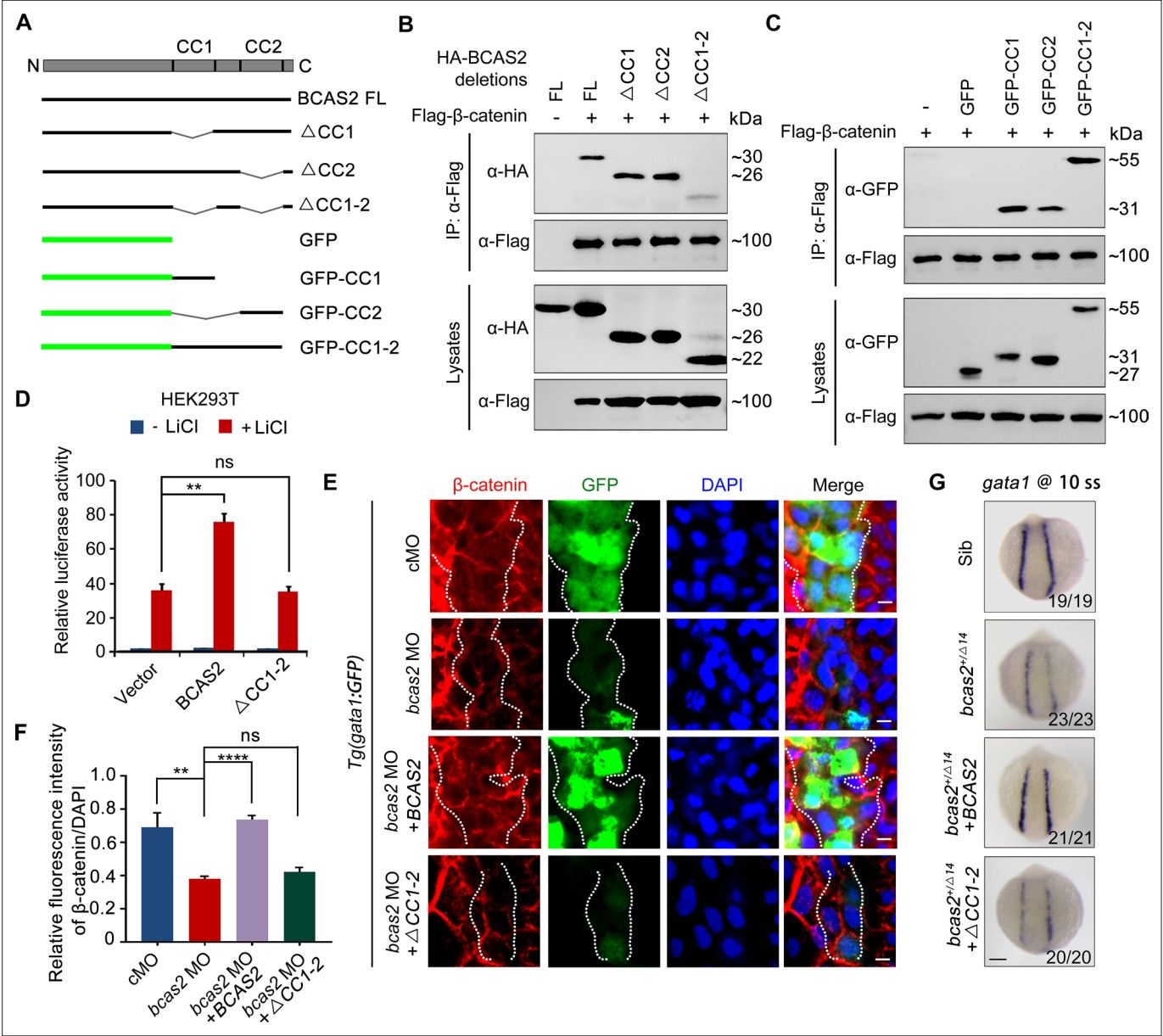

**Figure 7.** BCAS2 sequesters β-catenin in the nucleus via its CC domains. (**A**) Schematics of full length and deletion mutants of BCAS2. (**B, C**) HEK293T cells were transfected with Flag-β-catenin and indicated deletion mutants of BCAS2. Cell lysates were subjected to immunoprecipitation with anti-Flag antibody. Eluted proteins were immunoblotted using anti-HA (**B**) or anti-GFP antibodies (**C**) for BCAS2 detection. (**D**) HEK293T cells transfected with the indicated plasmids were treated with 100 ng/ml LiCl for 12 h, and then subjected to luciferase assay (n=3). ns, not significant; **p<0.01 (Student's *t*-test). (**E, F**) Immunofluorescence staining of β-catenin in *Tg(gata1:GFP)* embryos at 16 hpf. The embryos were injected with 8 ng *bcas2* MO and 300 pg of full-length *BCAS2* mRNA or *ΔCC1-2* mRNA at the one-cell stage. The relative fluorescence intensity of nuclear β-catenin was quantified in (**F**) (n=5). ns, not significant; **p<0.01; ****p<0.0001 (Student's *t*-test). (**G**) Transcripts of *gata1* were evaluated by WISH in *bcas2+/Δ14* embryos injected with 300 pg of *BCAS2* mRNA or *ΔCC1-2* mRNA. Scale bars, 5 μm (**E**), 100 μm (**G**).

The online version of this article includes the following source data and figure supplement(s) for figure 7:

**Source data 1.** Original western blots for *Figure 7B and C*, indicating the relevant bands and treatments.

**Source data 2.** Original files for western blot analysis in *Figure 7B and C*.

**Figure supplement 1.** Overexpression of the CC domains of BCAS2 restores nuclear β-catenin accumulation in *bcas2* morphants.

**Figure supplement 2.** Haploinsufficiency of *bcas2* does not affect pre-mRNA splicing during primitive hematopoiesis.

**Figure supplement 2—source data 1.** PDF file containing original gel images for *Figure 7—figure supplement 2B, C, D* with the relevant bands indicated.

**Figure supplement 2—source data 2.** Original gel images in *Figure 7—figure supplement 2B, C, D*.

## Cell lines and transfection

HEK293T (RRID:CVCL_0063), HeLa (RRID:CVCL_0030), SW480 (RRID:CVCL_0546), and L cells (RRID:CVCL_4536) were obtained from ATCC. All cell lines were authenticated by Short Tandem Repeat (STR) analysis, tested for mycoplasma contamination, and confirmed to be negative. *Bcas2*-cKO MEFs were prepared from *Bcas2^{F/F}* embryos at E13.5. Cells were cultured in Dulbecco's modified eagle's medium (HyClone) supplemented with 10% fetal bovine serum (HyClone) and 1% penicillin-streptomycin (HyClone) at 37°C and 5% CO_2. L cells expressing Wnt3a were maintained under similar conditions in the presence of 400 µg/ml G-418, from which Wnt3a conditioned medium (Wnt3a CM) was generated. Culture medium prepared from L cells was used as a control. To stimulate Wnt signaling, cells were treated with Wnt3a CM in a 1:1 ratio with normal media. To deplete *Bcas2* expression, *Bcas2*-cKO MEFs were cultured in medium containing 2 µM tamoxifen for 72 h and the knockout efficiency was evaluated using western blot analysis. The same cells cultured without tamoxifen were used as a control. To silence *BCAS2* expression, shRNA constructs in pLL 3.7-GFP plasmid were generated to target the following sequences: shRNA1, GAATGTGTAAACAATTCTA; shRNA2: GAAGGAACTTCAGAAGTTA. Transfection was performed with Lipofectamine 2000 (Invitrogen Cat# 11668019) according to the manufacturer's instructions.

## Generation of CRISPR-Cas9-mediated *bcas2* knockout zebrafish

The *bcas2* knockout zebrafish mutants were generated by CRISPR-Cas9 system as previously described (*Chang et al., 2013*). The guide RNA was designed to target the sequences 5'-GGCG CAGCTGGAGCATCAGG-3' within exon 4 of *bcas2*. Humanized Cas9 mRNA and gRNA were co-injected into wild-type embryos at the one-cell stage. Embryos or adult fin clips were collected to prepare genomic DNA. To screen for mutant alleles, the genomic regions containing gRNA-targeted sequences were amplified by polymerase chain reaction (PCR) with primers listed in *Supplementary file 1*. The PCR products were sequenced or digested with T7 endonuclease or restriction enzyme FspI for genotyping.

## RNA, morpholinos, and microinjection

Capped mRNAs for human *BCAS2*, *BCAS2 △CC1-2*, *BCAS2 CC1-2*, and mouse *ΔN-β-catenin* mRNA were synthesized from the corresponding linearized plasmids using an mMESSAGE mMACHINE T7 transcription kit (Ambion Cat# AM1344). Morpholino (MOs) were designed and purchased from Gene Tools: mismatch MO (cMO 5'-AGCCACTCATCCTGCTCCTCCCATC-3'), and *bcas2* translation-blocking MO (tMO; 5'-AGCGACTGATGCTGGTCCTGCCATC-3'). The mRNAs and morpholinos were injected into embryos at the 1- to 2 cell stage.

## Whole-mount in situ hybridization

Digoxigenin-labeled and fluorescein-labeled probes were synthesized using a RNA Labeling kit (Roche Cat# 11175025910). WISH and double FISH for zebrafish embryos were performed following previously published methods (*Jia et al., 2008*; *Welten et al., 2006*). Anti-digoxigenin-POD (Roche Cat# 11633716001) and anti-fluorescein-POD (Roche Cat# 11426346910) were used to detect digoxigenin-labeled probes and fluorescein-labeled probes, respectively. After WISH, the stained embryos were embedded in OCT and sections were prepared with a LEICA CM1900. The mouse yolk sac layers were separated as previously described (*Wallingford and Giachelli, 2014*).

## *o*-Dianisidine staining

To evaluate hemoglobin level, embryos were harvested at 36 hpf or 48 hpf, then stained with *o*-dianisidine as previously described (*Lieschke et al., 2001*).

## Proliferation and apoptosis assays

Embryos were incubated with 10 mM bromodeoxyuridine (BrdU) (Sigma-Aldrich Cat# B5002) for 20 min. The incorporated BrdU was detected with anti-BrdU (Sigma-Aldrich Cat# B2531, RRID:AB_476793) antibody. TUNEL staining was performed using In Situ Cell Death Detection Kit, TMR red (Roche Cat# 12156792910) according to the manufacturer's recommendation.

## Heat shock treatment

To induce *dkk1* expression, *Tg(hsp70l:dkk1b-GFP)* embryos were subjected to heat shock (42°C) for 10 min at 10 hpf, and then collected at the indicated stage for WISH.

## Dual reporter assay

HEK293T cells or MEFs were seeded in 24-well plates and transfected with a Super-TOPflash plasmid containing multimerized TCF-binding elements and a Renilla luciferase plasmid, along with the indicated vectors. Then cells were treated with 100 ng/ml LiCl and/or Wnt3a CM for 12 h and assayed for luciferase activity using the Dual luciferase system (Promega Cat# E1910).

## Immunoprecipitation, GST pulldown, and western blotting

For immunoprecipitation, HEK293T cells were transfected with the indicated plasmids and collected 48 h after transfection. Subsequently, HEK293T cells were lysed in a lysis buffer (10 mM Tris-HCl, pH 7.5, 150 mM NaCl, 2 mM EDTA, and 0.5% Nonidet P-40) containing protease inhibitors. Immunoprecipitation was performed in accordance with the standard protocols.

For GST pulldown assay, GST, GST tagged β-catenin ARM 1–12 and His tagged BCAS2 were expressed in *Escherichia coli* BL21, then purified using Glutathione-Sepharose 4B beads (GE Healthcare Cat# 71024800-GE) and HisPur Ni-NTA beads (Thermo Fisher Cat# 88831), respectively. GST and GST-β-catenin ARM 1–12 proteins were immobilized onto Glutathione-Sepharose 4B beads and incubated with purified His-BCAS2 at 4°C for 4 h. Beads were washed three times and analyzed using western blotting.

Cytoplasmic and nuclear extracts were separated with nuclear and cytoplasmic extraction kit (CWBIO Cat# CW0199). Cell lysates were subjected to immunoprecipitation with anti-Flag M2 affinity gel (Sigma-Aldrich Cat# A2220, RRID:AB_10063035) or anti-c-Myc agarose affinity gel (Sigma-Aldrich Cat# A7470, RRID:AB_10109522) antibodies. Proteins were analyzed by western blot using anti-Flag (Sigma-Aldrich Cat# F2555, RRID:AB_796202), anti-HA (CWBIO Cat# CW0092A), anti-β-catenin (Abmart Cat# M24002, RRID:AB_2920853), anti-BCAS2 (Proteintech Cat# 10414–1-AP, RRID:AB_2063400), anti-β-Tubulin (CWBIO Cat# CW0265A), anti-GFP (Thermo Fisher Scientific Cat# A-11120, RRID:AB_221568), anti-Histone H3 (Abcam Cat# ab1791, RRID:AB_302613), anti-GST (Sigma-Aldrich Cat# SAB4200237, RRID:AB_2858197), and anti-His Tag (Beyotime Cat# AF5060) antibodies.

## Immunofluorescence staining

Cells on coverslips and embryos were processed for immunofluorescence staining as previously described (*Wei et al., 2017*; *Yang et al., 2022*). Before fixation, *bcas2*-deficient MEFs were treated with a concentration of 20 µM MG132 or 20 nM LMB for 6 h, while *Tg(gata1:GFP)* embryos were treated with 20 nM LMB from the bud stage to the 10-somite stage. The prepared samples were stained with anti-BCAS2 (Proteintech Cat# 10414-1-AP, RRID:AB_2063400), anti-β-catenin (Abmart Cat# M24002, RRID:AB_2920853), and anti-GFP (Thermo Fisher Scientific Cat# A-11122, RRID:AB_221569) antibodies. Meanwhile, 4',6-Diamidine-2'-phenylindole dihydrochloride (DAPI, Sigma-Aldrich Cat# 10236276001) was used to label nuclei. Fluorescence imaging was performed using a Nikon A1R Confocal Laser Scanning Microscope (RRID:SCR_020317), and all images were captured with the same settings. The relative fluorescence intensity was calculated by dividing the fluorescence intensity of nuclear β-catenin by the fluorescence intensity of DAPI.

## Bimolecular fluorescence complementation assay

To construct the plasmids for BiFC, BCAS2 was fused to the N-terminal half of yellow fluorescent protein (YN-BCAS2) and β-catenin to the C-terminal half (YC-β-catenin). YN-BCAS2 and YC-β-catenin were either individually or collectively transfected into HeLa cells. Fluorescence was detected 48 h after transfection using a Nikon A1R Confocal Laser Scanning Microscope (RRID:SCR_020317).

## Fluorescence recovery after photobleaching

BCAS2 and GFP tagged S37A-β-catenin were co-transfected into HeLa cells. Fluorescence recovery after photobleaching (FRAP) assay was performed according to previously reported methods (*Schmierer and Hill, 2005*). The cells were bleached by the 488 nm laser line of the 20 mW argon

laser at 100% power. About 90% of nuclear or cytoplasmic GFP signal was bleached. Images were acquired with 35 frames at 25 s intervals by a Zeiss LSM 510 Confocal Microscope (RRID:SCR_018062).

### RNA sequencing

Embryos were collected at the 10-somite stage and gently transferred into lysis buffer. Reverse transcription was performed using a SMARTer Ultra Low RNA Kit (Clontech Cat# 634437) directly from the cell lysates. The cDNA library was prepared using an Advantage 2 PCR Kit (Clontech Cat# 639206) and then sequenced via the Illumina NovaSeq 6000 Sequencing System (RRID:SCR_016387). The difference in the number of alternative splicing events between groups was analyzed using rMATS (RRID:SCR_023485, version 4.1.0).

### Reverse transcription PCR

Total RNA was isolated from wild-type and *bcas2* mutant embryos at the 10-somite stage with Micro-Elute Total RNA kit (OMEGA Cat# R6831-01), followed by reverse transcription using ReverTra Ace qPCR RT Kit (Toyobo Cat# FsQ-101). The cDNA was amplified with the primers listed in *Supplementary file 2*.

### Quantification and statistical analysis

Images were quantified with ImageJ (RRID:SCR_003070). Statistical data were analyzed using GraphPad Prism (RRID:SCR_002798). Comparisons between experimental groups were done using the Student's *t*-test. Data are presented as mean ± SD. $p<0.05$, $p<0.01$, $p<0.001$, and $p<0.0001$ were considered statistically significant and marked with *, **, ***, and ****, respectively (Student's *t*-test).

### Materials availability statement

Further information and requests for reagents should be directed to the corresponding author, Qiang Wang (qiangwang@scut.edu.cn).

## Acknowledgements

We acknowledge the financial support of the National Natural Science Foundation of China (32025014 and 32330029) and the National Key Research and Development Program of China (2018YFA0800200 and 2020YFA0804000).

## Additional information

### Funding

| Funder | Grant reference number | Author |
| --- | --- | --- |
| National Natural Science Foundation of China | 32025014 | Qiang Wang |
| National Natural Science Foundation of China | 32330029 | Qiang Wang |
| National Key Research and Development Program of China | 2018YFA0800200 | Qiang Wang |
| National Key Research and Development Program of China | 2020YFA0804000 | Qiang Wang |

The funders had no role in study design, data collection and interpretation, or the decision to submit the work for publication.

### Author contributions

Guozhu Ning, Conceptualization, Data curation, Software, Formal analysis, Investigation, Visualization, Methodology, Writing – original draft; Yu Lin, Data curation, Software, Formal analysis, Investigation,

Visualization, Methodology; Haixia Ma, Jiaqi Zhang, Liping Yang, Zhengyu Liu, Validation, Methodology; Lei Li, Resources; Xinyu He, Writing – review and editing; Qiang Wang, Supervision, Funding acquisition, Project administration

### Author ORCIDs
Guozhu Ning ⓘ https://orcid.org/0000-0003-3392-3007
Lei Li ⓘ https://orcid.org/0000-0001-5478-5681
Xinyu He ⓘ https://orcid.org/0009-0006-0829-9125
Qiang Wang ⓘ https://orcid.org/0000-0002-8735-8771

### Ethics
Our studies including animal maintenance and experiments were performed in compliance with the guidelines of the Animal Care and Use Committee of the South China University of Technology (Permission Number: 2023092).

Reviewer #1 (Public review): https://doi.org/10.7554/eLife.100497.3.sa1
Reviewer #2 (Public review): https://doi.org/10.7554/eLife.100497.3.sa2
Reviewer #3 (Public review): https://doi.org/10.7554/eLife.100497.3.sa3
Author response https://doi.org/10.7554/eLife.100497.3.sa4

## Additional files

### Supplementary files
MDAR checklist

Supplementary file 1. Primers used for genotyping.

Supplementary file 2. Primers used for reverse transcription-PCR.

### Data availability
All data generated or analyzed during this study are included in the manuscript and/or supplementary materials. RNA sequencing data have been deposited in GEO under accession codes GSE297155. Original western blot images have been provided as source data.

The following dataset was generated:

| Author(s) | Year | Dataset title | Dataset URL | Database and Identifier |
| --- | --- | --- | --- | --- |
| Ning G, Lin Y | 2025 | BCAS2 promotes primitive hematopoiesis by sequestering β-catenin within the nucleus | https://www.ncbi.nlm.nih.gov/geo/query/acc.cgi?acc=GSE297155 | NCBI Gene Expression Omnibus, GSE297155 |

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
