## [Editor Report · eLife Assessment]

This **important** work supports the role of breast carcinoma amplified sequence 2 (Bcas2) in positively regulating primitive wave hematopoiesis through amplification of beta-catenin-dependent (canonical) Wnt signaling. The study is **convincing**: it uses appropriate and validated methodology in line with the current state-of-the-art, and there is a first-rate analysis of a strong phenotype with highly supportive mechanistic data. The findings shed light on the controversial question of whether, when, and how canonical Wnt signaling may be involved in hematopoietic development. The work will be of interest to hematologists and developmental biologists.

---

## [Referee Report · Reviewer #1 (Public review)]

Summary:

In this manuscript, Ning et al. reported that Bcas2 played an indispensable role in zebrafish primitive hematopoiesis via sequestering β-catenin in the nucleus. The authors showed that loss of Bcas2 caused primitive hematopoietic defects in zebrafish. They unraveled that Bcas2 deficiency promoted β-catenin nuclear export via a CRM1-dependent manner in vivo and in vitro. They further validated that BCAS2 directly interacted with β-catenin in the nucleus and enhanced β-catenin accumulation through its CC domains. They unveil a novel insight into Bcas2, which is critical for zebrafish primitive hematopoiesis via regulating nuclear β-catenin stabilization rather than its canonical pre-mRNA splicing functions. Overall, the study is impressive and well-performed, although there are also some issues to address.

Strengths:

The study unveils a novel function of Bcas2, which is critical for zebrafish primitive hematopoiesis by sequestering β-catenin. The authors validated the results in vivo and in vitro. Most of the figures are clear and convincing. This study nicely complements the function of Bcas2 in primitive hematopoiesis.

Comments on revisions:

The authors have nicely answered all my questions, I have no problem.

---

## [Referee Report · Reviewer #2 (Public review)]

Summary:

Ning and colleagues present studies supporting a role for breast carcinoma amplified sequence 2 (Bcas2) in positively regulating primitive wave hematopoiesis through amplification of beta-catenin-dependent (canonical) Wnt signaling. The authors present compelling evidence that zebrafish bcas2 is expressed at the right time and place to be involved in primitive hematopoiesis, that there are primitive hematopoietic defects in hetero- and homozygous mutant and knockdown embryos, that Bcas2 mechanistically positively regulates canonical Wnt signaling, and that Bcas2 is required for nuclear retention of B-cat through physical interaction involving armadillo repeats 9-12 of B-cat and the coiled-coil domains of Bcas2. Overall, the data and writing are clean, clear, and compelling. This study is a first rate analysis of a strong phenotype with highly supportive mechanistic data. The findings shed light on the controversial question of whether, when, and how canonical Wnt signaling may be involved in hematopoietic development.

In the revised version of their previous work, they have included responses to some of our suggestions for minor experiments and edits. We previously suggested they examine the structural compatibility of a Bcas2/beta-catenin dimer with binding to the DNA-binding protein Tcf7l1 (previously Tcf3), which would be expected for a beta-catenin nuclear-retention factor that potentiates canonical Wnt signaling responses. Although the authors did not test compatibility of Bcas2 with Tcf3 binding to beta-catenin, they show that a three-way complex with the family member Tcf4 is possible (Fig. S12), which suggests that Lef/Tcf family binding in general is plausible.

The authors' acceptance of our suggestion to evaluate cdx and hox gene expression is welcome, as these genes have previously been defined as canonical Wnt targets (Lengerke et al., 2009) that regionalize the lateral plate mesoderm (LPM) and confer pre-hematopoietic identity there (Davidson et al., 2003; Davidson and Zon, 2004). The authors' finding that cdx4 and hoxa9a are diminished in the bcas2 mutants (Fig. S7) validates this suggestion and seem to imply that the primary defect here is specification of the early hematopoietic field in the LPM, however the results are a little confusing or surprising given that scl - which is unaffected in the bcas2 mutant (Fig. 2A) - is a downstream target of Cdx4 (Davidson et al., 2003, Fig. 1b, 3d). The results in the current submission imply that early maintenance of pre-hematopoietic competence in the LPM is a canonical-Wnt-directed phenomenon separable from the earliest specification of the hematopoietic field. We believe it would be of value to further evaluate regulation of cdx1, which has been shown to cooperate with cdx4 in regulation of the LPM hematopoietic field, as well as analyze some of the putative downstream hox family targets.

We previously reviewed the article as suitable for publication and we continue to support our prior assessment. The authors have presented strong data supporting a role for Bcas2 in hematopoietic development across phyla and a mechanistic involvement in promoting canonical Wnt signaling.

Strengths:

(1) The study features clear and compelling phenotypes and results.

(2) The manuscript narrative exposition and writing are clear and compelling.

(3) The authors have attended to important technical nuances sometimes overlooked, for example, focusing on different pools of cytosolic or nuclear b-catenin.

(4) The study sheds light on a controversial subject: regulation of hematopoietic development by canonical Wnt signaling and presents clear evidence of a role.

(5) The authors present evidence of phylogenetic conservation of the pathway.

---

## [Referee Report · Reviewer #3 (Public review)]

Summary:

This manuscript utilized zebrafish bcas2 mutants to study the role of bcas2 in primitive hematopoiesis, and further confirms that it has a similar function in mice. Moreover, they showed that bcas2 regulates the transition of hematopoietic differentiation from angioblasts via activating Wnt signaling. By performing a series of biochemical experiments, they also showed that bcas2 accomplishes this by sequestering b-catenin within the nucleus, rather than through its known function in pre-mRNA splicing.

Strengths:

The work is well-performed, and the manuscript is well-written.

Comments on revisions:

The revised manuscript is substantially improved, and all my previous questions are now well addressed.

---

## [Author Response]

The following is the authors’ response to the original reviews

**Reviewer #1 (Public Review):**
Summary:In this manuscript, Ning et al. reported that Bcas2 played an indispensable role in zebrafish primitive hematopoiesis via sequestering β-catenin in the nucleus. The authors showed that loss of Bcas2 caused primitive hematopoietic defects in zebrafish. They unraveled that Bcas2 deficiency promoted β-catenin nuclear export via a CRM1-dependent manner in vivo and in vitro. They further validated that BCAS2 directly interacted with β-catenin in the nucleus and enhanced β-catenin accumulation through its CC domains. They unveil a novel insight into Bcas2, which is critical for zebrafish primitive hematopoiesis via regulating nuclear β-catenin stabilization rather than its canonical pre-mRNA splicing functions. Overall, the study is impressive and well-performed, although there are also some issues to address.Strengths:The study unveils a novel function of Bcas2, which is critical for zebrafish primitive hematopoiesis by sequestering β-catenin. The authors validated the results in vivo and in vitro. Most of the figures are clear and convincing. This study nicely complements the function of Bcas2 in primitive hematopoiesis.Weaknesses:A portion of the figures were over-exposed.

Thank you for the time reviewing our manuscript. We agree with your suggestion and the exposure of Figure 5C and Figure 7E has been reduced. We hope that the revisions will meet your expectation.

**Reviewer #2 (Public Review):**
Summary:Ning and colleagues present studies supporting a role for breast carcinoma amplified sequence 2 (Bcas2) in positively regulating primitive wave hematopoiesis through amplification of beta-catenin-dependent (canonical) Wnt signaling. The authors present compelling evidence that zebrafish bcas2 is expressed at the right time and place to be involved in primitive hematopoiesis, that there are primitive hematopoietic defects in hetero- and homozygous mutant and knockdown embryos, that Bcas2 mechanistically positively regulates canonical Wnt signaling, and that Bcas2 is required for nuclear retention of B-cat through physical interaction involving armadillo repeats 9-12 of B-cat and the coiled-coil domains of Bcas2. Overall, the data and writing are clean, clear, and compelling. This study is a first-rate analysis of a strong phenotype with highly supportive mechanistic data. The findings shed light on the controversial question of whether, when, and how canonical Wnt signaling may be involved in hematopoietic development. We detail some minor concerns and questions below, which if answered, we believe would strengthen the overall story and resolve some puzzling features of the phenotype. Notwithstanding these minor concerns, we believe this is an exceptionally well-executed and interesting manuscript that is likely suitable for publication with minor additional experimental detail and commentary.Strengths:(1) The study features clear and compelling phenotypes and results.(2) The manuscript narrative exposition and writing are clear and compelling.(3) The authors have attended to important technical nuances sometimes overlooked, for example, focusing on different pools of cytosolic or nuclear b-catenin.(4) The study sheds light on a controversial subject: regulation of hematopoietic development by canonical Wnt signaling and presents clear evidence of a role.(5) The authors present evidence of phylogenetic conservation of the pathway.Weaknesses:(1) The authors present compelling data that Bcas2 regulates nuclear retention of B-cat through physical association involving binding between the Bcas2 CC domains and B-cat arm repeats 9-12. Transcriptional activation of Wnt target genes by B-cat requires physical association between B-cat and Tcf/Lef family DNA binding factors involving key interactions in Arm repeats 2-9 (Graham et al., Cell 2000). Mutually exclusive binding by B-cat regulatory factors, such as ICAT that prevent Tcf-binding is a documented mechanism (e.g. Graham et al., Mol Cell 2002). It would appear - based on the arm repeat usage by Bcas2 (repeats 9-12)-that Bcas2 and Tcf binding might not be mutually exclusive, which would support their model that Bcas2 physical association with B-cat to retain it in the nucleus would be compatible with co-activation of genes by allowing association with Tcf. It might be nice to attempt a three-way co-IP of these factors showing that B-cat can still bind Tcf in the presence of Bcas2, or at least speculate on the plausibility of the three-way interaction.

We appreciate your assessment and generous comments for the manuscript. As you mentioned, the binding sites for TCF on β-catenin almost do not overlap with those for BCAS2. It is likely that BCAS2-mediated nuclear sequestration of β-catenin would be compatible with the initiation of gene transcription by allowing TCF to associate with β-catenin. To test this possibility, we have taken your suggestion and performed co-IP assays. The results showed that β-catenin still bound with TCF4 in the presence of BCAS2 (Supplemental Figure 12), confirming that the binding of BCAS2 to β-catenin would not interfere with the formation of β-catenin/TCF complex.

(2) A major way that canonical Wnt signaling regulates hematopoietic development is through regulation of the LPM hematopoietic competence territories by activating expression of cdx1a, cdx4, and their downstream targets hoxb5a and hoxa9a (Davidson et al., Nature 2003; Davidson et al., Dev Biol 2006; Pilon et al., Dev Biol 2006; Wang et al., PNAS 2008). Could the authors assess (in situ) the expression of cdx1a, cdx4, hoxb5a, and hoxa9a in the bcas2 mutants?

We agree with your suggestion and have examined the expression of *cdx4* and *hoxa9a* by performing WISH. Diminished expression of *cdx4* and *hoxa9a* was detected in the lateral plate mesoderm of *bcas2*^+/-^ embryos at the 6-somite stage (Supplemental Figure 7).

(3) The authors show compellingly that even heterozygous loss of bcas2 has strong Wnt-inhibitory effects. If Bcas2 is required for canonical Wnt signaling and bcas2 is expressed ubiquitously from the 1-cell stage through at least the beginning of gastrulation, why do bcas2 KO embryos not have morphological axis specification defects consistent with loss of early Wnt signaling, like loss of head (early), or brain anteriorization (later)? Could the authors provide some comments on this puzzle? Or if they do see any canonical Wnt signaling patterning defects in het- or homozygous embryos, could they describe and/or present them?

You have raised an interesting question. In fact, we did not observe ventralization or axis determination defects in the early embryos of *bcas2*^+/-^ mutants. Even in the very small number of homozygous mutant embryos, we did not find such morphological defects. Given that the homozygous and heterozygous mutant embryos were derived from crossing *bcas2*^+/-^ males with *bcas2*^+/-^ females, maternal Bcas2 might still remain and function in these embryos during gastrulation when axis determination and neural patterning took place. Accordingly, we have expanded our discussion to incorporate these insights (Line 565-572).

**Reviewer #3 (Public Review):**
Summary:This manuscript utilized zebrafish bcas2 mutants to study the role of bcas2 in primitive hematopoiesis and further confirms that it has a similar function in mice. Moreover, they showed that bcas2 regulates the transition of hematopoietic differentiation from angioblasts via activating Wnt signaling. By performing a series of biochemical experiments, they also showed that bcas2 accomplishes this by sequestering b-catenin within the nucleus, rather than through its known function in pre-mRNA splicing.Strengths:The work is well-performed, and the manuscript is well-written.Weaknesses:Several issues need to be clarified.(1) Is wnt signaling also required during hematopoietic differentiation from angioblasts? Can the authors test angioblast and endothelial markers in embryos with wnt inhibition? Also, can the authors add export inhibitor LMB to the mouse mutants to test if sequestering of b-catenin by bcas2 is conserved during primitive hematopoiesis in mice?

Thank you very much for your appreciation and detailed assessment. To test whether Wnt signaling is also required during hematopoietic differentiation from angioblasts, wild-type embryos were exposed to 10 µM CCT036477, a small molecule β-catenin antagonist, from 9 hpf and then collected for WISH experiments. As shown in Supplemental Figure 8, the expression of hemangioblast markers *npas4l*, *scl*, and *gata2* and endothelial marker *fli1a* remained unchanged, but the expression of erythroid progenitor marker *gata1* was significantly reduced. These results suggest that canonical Wnt pathway may not be required for the generation of hemangioblasts or their endothelial differentiation, but is pivotal for their hematopoietic differentiation.

It is quite difficult to validate the conserve role of BCAS2 during primitive hematopoiesis in mice, because the toxicity of LMB may cause severe adverse effects in mice.[1,2]

(2) Bcas2 is required for primitive myelopoiesis in ALM. Does bcas2 play a similar function in primitive myelopoiesis, or is bcas2/b-catenin interaction more important for hematopoietic differentiation in PLM?

You have raised an important question. In our study, we have demonstrated that the expression of myeloid progenitor marker *pu.1* was significantly decreased in *bcas2* mutants, hinting that Bcas2 is pivotal for primitive myelopoiesis. To further clarify the function of Bcas2 in primitive myelopoiesis, we injected 8 ng of *bcas2* morpholino into *Tg(coro1a:GFP)* embryos at the 1-cell stage and examined β-catenin distribution at 17 hpf via immunostaining. We observed a significant decline of nuclear β-catenin in primitive myeloid cells (Supplemental Figure 9), indicating that Bcas2 is highly likely to play a similar role in sequestering β-catenin within the nucleus during primitive myelopoiesis.

(3) Is it possible that CC1-2 fragment sequester b-catenin? The different phenotypes between this manuscript and the previous article (Yu, 2019) may be due to different mutations in bcas2. Is it possible that the bcas2 mutation in Yu's article produces a complete CC1-2 fragment, which might sequester b-catenin?

This is an interesting perspective. To test the possibility that CC1-2 sequesters β-catenin, mRNA expressing the CC domains of BCAS2 has been co-injected with *bcas2* morpholino into *Tg(gata1:GFP)* embryo at the one-cell stage. Increased nuclear β-catenin levels were detected in the GFP-positive hematopoietic progenitor cells at 16 hpf (Supplemental Figure 11). Our findings support that CC1-2 fragment of BCAS2 can sequester β-catenin within the nucleus.

In the previous article (Yu, 2019), a deletion 5 bases mutation in the third exon of BCAS2 was produced by TALEN, therefore the CC domains of this mutant should be affected. It is difficult to conclude that the mutant BCAS2 protein in Yu’s study still remains association with β-catenin.

(4) Can the author clarify what embryos the arrows point to in SI Figure 2D? In SI Figure 6B and B', can the author clarify how the nucleus and cytoplasm are bleached? In B, the nucleus also appears to be bleached.

Thank you for your query and suggestion. In our revisions, the corresponding clarifications have been supplemented (Line 239-242; Line 978-979).

We acknowledge that the nuclei in both the BCAS2 overexpression group and control group were slightly bleached. Given that we have performed real-time analysis for fluorescent recovery after photobleaching, and we have observed a much slower recovery of cytoplasmic fluorescence in BCAS2 overexpressed cells, the conclusion that BCAS2 inhibits the nuclear export of β-catenin but not its nuclear import, remains changed.

**Reviewer #1 (Recommendations For The Authors):**
Major concerns:(1) In this study, the authors detected β-catenin distribution in erythrocytes (gata1-GFP+ cells). Estimating the β-catenin distribution in the myeloid cells is recommended.

Thank you for your assessment and we have taken your suggestion. *Tg(coro1a:GFP)* embryos, which is commonly used to track both macrophages and neutrophils,[3] were injected with 8 ng of *bcas2* morpholino into at the 1-cell stage and collected for immunostaining to examine the β-catenin distribution at 17 hpf. We observed a significant decline of nuclear β-catenin in primitive myeloid cells (Supplemental Figure 9). This result indicates that Bcas2 is highly likely to play a similar role in sequestering β-catenin within the nucleus during primitive myelopoiesis.

(2) The reduced nuclear localization of β-catenin in Figure 3H required further evidence. It would be helpful if the authors quantified the fluorescence intensity in the cell nucleus and cytoplasm. Meanwhile, the figures (Figure 5C, Figure 7E) were over-exposed. Please validate these figures.

Thank you for your suggestions. We agree with you that the fluorescence intensity of β-catenin in the nucleus and cytoplasm should be quantified. However, as the nucleus comprises a large part of the cell, we believe it would be more appropriate to quantify the relative fluorescence intensity by dividing the fluorescence intensity of nuclear β-catenin by the fluorescence intensity of DAPI.

Such quantifications have been added for Figure 3G, 5C, 7E, S9A, and S13A. In addition, we have reduced the exposure of Figure 5C and Figure 7E. We hope that you will be satisfied with the revisions.

(3) The authors used cKO mice to validate that the erythrocytes were eliminated. It would be interesting to detect β-catenin distribution by immunofluorescent staining in primitive hematopoietic cells in cKO mice. Addressing this issue can provide further evidence to support the conservation of Bcas2.

We appreciate your suggestion. However, we found that red blood cells were almost eliminated in the yolk sac of *Bcas2F/F;Flk1-Cre* mice at E12.5. It is difficult to further detect β-catenin distribution in primitive erythroid cells in these mice.

(4) The authors discovered that Bcas2 mediated β-catenin nuclear export in a CRM1-dependent manner. CRM1 is a key regulator involved in the majority of factors of nuclear export via recognizing specific nuclear export signals (NES). Validating the NES of Bcas2 is recommended. Furthermore, I wonder about the relationship between Bcas2 and CRM1 in regulating β-catenin nuclear export. One possibility is that Bcas2 covers the NES to inhibit the interaction between CRM1 and β-catenin, thus leading to β-catenin accumulation in the cell nucleus. The authors should discuss this possibility accordingly.

Thank you for providing an interesting perspective. CRM1-mediated nuclear export of β-catenin usually requires CRM1 recognition and binding with the NES sequences in chaperon proteins, such as APC, Axin and Chibby.[4-6] Moreover, CRM1 can bind directly to and function as an efficient nuclear exporter for β-catenin.[7] Since BCAS2 has not been reported to contain any recognizable NES sequences, it will be interesting to investigate whether BCAS2 competitively inhibits β-catenin from associating with CRM1, or with the chaperone proteins. We have rewritten the discussion on CRM1-dependent nuclear export of β-catenin in line with your comments (Line 572-578).

(5) It would be interesting if the authors could answer the specificity in Bcas2-mediated protein nuclear export pathway. The authors should detect other classical factors (CRM1 mediated) distribution when loss of Bcas2.

Thank you for bringing up this point. To test whether BCAS2 specifically regulates CRM1-mediated nuclear export of β-catenin, we have investigated the nucleocytoplasmic distribution of other known CRM1 cargoes, such as ATG3 and CDC37L.[8] BCAS2 overexpression in HeLa cells slightly enhanced the nuclear localization of CDC37L, and had no significant impact on that of ATG3 (Supplemental Figure 11), indicating the specificity of BCAS2 in the regulation of CRM1-dependent nuclear export of β-catenin.

Minor concerns:(1) The name "bcas2Δ7+/- and bcas2Δ14+/-" should be changed into "bcas2+/Δ7 and bcas2+/Δ14"(+/Δ7 or +/Δ14 should be superior on the right).

Thank you for your suggestion. We have changed the names of the mutants throughout the manuscript.

(2) The scale bar position in the figures should be unified.

We agree with your suggestion and have unified the scale bar position in all figures.

(3) In Figure 4E, "Nuclear" should be changed into "Nucleus".

We apologize for the mistake and Figure 4E has been revised.

(4) There are some unaesthetic issues in the figures. The figures need to be further edited. Figure 3H "β-catenin and Merge", Figure 4D "Merge". All these words should be centered in the figures.

Thank you. We have edited all the figures to ensure that the text is centered.

**Reviewer #2 (Recommendations For The Authors):**
(1) It would be nice to have whole blot images for the Westerns in Supplementary Info.

Thank you for your suggestion. Whole images for immunoblotting have been supplemented as Source data.

(2) Line 292 change 5 hpf to 5 dpf.(3) Line 301 change "primary" to "primitive"?

We apologize for the mistakes. We have incorporated these suggestions in the revised manuscript and reexamined spelling throughout the paper.

(4) Figure S2C: is "Maker" a typographical error? Change to "ladder"?

We apologize for this typographical error and we have revised it in Figure S2C.

Reference

(1) Ishizawa J, Kojima K, Hail N, Tabe Y, Andreeff M. Expression, function, and targeting of the nuclear exporter chromosome region maintenance 1 (CRM1) protein. Pharmacology & Therapeutics. 2015;153:25-35.

(2) Li X, Feng Y, Yan MF, et al. Inhibition of Autism-Related Crm1 Disrupts Mitosis and Induces Apoptosis of the Cortical Neural Progenitors. Cerebral Cortex. 2020;30(7):3960-3976.

(3) Li L, Yan B, Shi YQ, Zhang WQ, Wen ZL. Live Imaging Reveals Differing Roles of Macrophages and Neutrophils during Zebrafish Tail Fin Regeneration. Journal of Biological Chemistry. 2012;287(30):25353-25360.

(4) Neufeld KL, Nix DA, Bogerd H, et al. Adenomatous polyposis coli protein contains two nuclear export signals and shuttles between the nucleus and cytoplasm. Proceedings of the National Academy of Sciences of the United States of America. 2000;97(22):12085-12090.

(5) Li FQ, Mofunanya A, Harris K, Takemaru KI. Chibby cooperates with 14-3-3 to regulate β-catenin subcellular distribution and signaling activity. Journal of Cell Biology. 2008;181(7):1141-1154.

(6) Cong F, Varmus H. Nuclear-cytoplasmic shuttling of Axin regulates subcellular localization of β-catenin. Proceedings of the National Academy of Sciences of the United States of America. 2004;101(9):2882-2887.

(7) Ki H, Oh M, Chung SW, Kim K. β-Catenin can bind directly to CRM1 independently of adenomatous polyposis coli, which affects its nuclear localization and LEF-1/β-catenin-dependent gene expression. Cell Biology International. 2008;32(4):394-400.

(8) Kirli K, Karaca S, Dehne HJ, et al. A deep proteomics perspective on CRM1-mediated nuclear export and nucleocytoplasmic partitioning. Elife. 2015;4.